# Structure and Biological Activity of Ergostane-Type Steroids from Fungi

**DOI:** 10.3390/molecules27072103

**Published:** 2022-03-24

**Authors:** Vladimir N. Zhabinskii, Pavel Drasar, Vladimir A. Khripach

**Affiliations:** 1Institute of Bioorganic Chemistry, National Academy of Sciences of Belarus, Kuprevich Str., 5/2, 220141 Minsk, Belarus; khripach@iboch.by; 2Department of Chemistry of Natural Compounds, University of Chemistry and Technology, Technicka 5, CZ-166 28 Prague, Czech Republic; pavel.drasar@vscht.cz

**Keywords:** ergosterol, ergosteroids, fungi, mushrooms, anticancer, antiviral, cytotoxicity

## Abstract

Mushrooms are known not only for their taste but also for beneficial effects on health attributed to plethora of constituents. All mushrooms belong to the kingdom of fungi, which also includes yeasts and molds. Each year, hundreds of new metabolites of the main fungal sterol, ergosterol, are isolated from fungal sources. As a rule, further testing is carried out for their biological effects, and many of the isolated compounds exhibit one or another activity. This study aims to review recent literature (mainly over the past 10 years, selected older works are discussed for consistency purposes) on the structures and bioactivities of fungal metabolites of ergosterol. The review is not exhaustive in its coverage of structures found in fungi. Rather, it focuses solely on discussing compounds that have shown some biological activity with potential pharmacological utility.

## 1. Introduction

Fungi are a rich source of chemical compounds with a wide spectrum of biological activity [1]. To survive in the environment in which they exist, they need to protect themselves from fungal infections. Therefore, it is not surprising that antimicrobial or antiviral compounds beneficial to humans can be isolated from many fungi [2]. A large number of currently used drugs have their origins in fungi [3]. Steroids occupy an important place among fungal constituents. The vast majority of them are ergosterol metabolites. The latter is the main sterol of fungi involved in the regulation of membrane fluidity and structure as well as performing immunological functions [4]. Fungal ergosterol derivatives are often referred to as “ergostane-type steroids” [5,6,7,8,9,10,11,12] or “ergosteroids” [13,14,15,16,17]. One should bear in mind, however, that the application of the term “ergosteroids” can be confusing, as it was also suggested by Lardy et al. [18] to structurally different dehydroepiandrosterone derivatives based on their mode of action (influence on energy metabolism).

Ergostane-type steroids are characteristic not only of fungi but also of plants [19,20,21] and sponges [22]. These steroids are not a focus of the present paper. The purpose of this review is to highlight current knowledge on the structures and biological activities of fungal constituents, built on an ergostane skeleton **1** (Figure 1) or structures of which can be traced back to it. Currently, there are a number of reviews in this area dedicated to certain aspects or groups of ergostanes. A nice review on chemistry, biology, and medicinal aspects of rearranged ergostane-type natural products has been published recently by Heretsch et al. [23]. A detailed literature survey by Merdivan and Lindequist was dedicated to the consideration of biological activities of a single compound (ergosterol 5α,8α-endoperoxide) [24]. Many reviews discuss ergostane-type steroids as a part of fungal compositional diversity constituents [25,26,27,28,29,30,31,32].

## 2. Sterols

### 2.1. Ergosterol

Detailed studies of the biological effects of fungi have shown that some of them can be attributed to ergosterol (**2**) [33,34,35,36,37,38]. That is why ergosterol itself has attracted considerable attention as a potential lead for the development of new therapeutics. Its anticancer properties were investigated on the lungs [39], liver [40,41], breast [42], human gastric [43], and prostate [44] cancer cell lines.

Ergosterol treatment of mice inoculated with breast cancer cells prolonged mouse survival [42]. Suppression of cancer cell viability was explained by apoptosis and by up-regulating Foxo3 and Foxo3 downstream molecules Bim, Fas, and Fas L. 

The antitumor potential of ergosterol was studied upon its application with amphotericin B [40]. The latter is a macrolide antifungal agent that is also used to reverse chemotherapeutic drug resistance. The combined treatment of liver cancer cell lines with ergosterol followed by amphotericin B resulted in a significant decrease of their viability as a result of necrotic cell death. 

Experiments on reversing multidrug resistance in cancer cells were also performed using drug-sensitive human gastric carcinoma cell line SGC7901 and its adriamycin-resistant counterpart SGC7901/Adr. Ergosterol at concentrations below 5 μM has been shown to enhance the cytotoxicity of adriamycin on SGC7901/Adr cells [43]. 

In experiments with Hep2 cancer cells, it was shown that ergosterol inhibited cell growth with IC_50_ value of 40 μM/mL [41]. The observed effect was explained by the pro-oxidant properties of ergosterol on the Hep2 cells. 

Different effects have been noted for androgen-dependent LNCaP and androgen-independent DU-145 prostate cancer cells [44]. While ergosterol exerted an antiproliferative action on LNCaP, it promoted cell proliferation on DU-145. The authors [44] suggested that the observed difference may be related to the ability of ergosterol to act as a ligand for the androgen receptor. 

Experiments with rats fed with a diet containing 0.1% ergosterol have shown a certain bladder carcinogenesis-preventing effect [45]. It was supposed that the observed effect is due to an androgen receptor expression-reducing action of brassicasterol (metabolite of ergosterol) on bladder epithelial cells. 

Several studies have reported the anti-inflammatory effects of ergosterol. Its treatment of RAW 264.7 macrophages inhibited lipopolysaccharide-induced inflammation by suppressing the production of tumor necrosis factor-α and expression of cyclooxygenase-2 [46]. The inhibitory effect of ergosterol on degranulation of mucosal-type murine bone marrow-derived mast cells [47] or basophilic leukemia (RBL-2H3) cells [48] was associated with inhibition of β-hexosaminidase and histamine release in antigen-stimulated cells and was of interest for the treatment of allergic diseases dependent on mast cells. 

Pretreatment of mice with ergosterol at doses of 25 and 50 mg/kg reduced lipopolysaccharide-induced histopathological changes in the lungs [49]. In addition, inhibition of inflammatory cells and pro-inflammatory cytokines, including tumor necrosis factor-α and interleukin-6, was observed. Similar effects were found on cigarette smoke-induced chronic obstructive pulmonary disease (COPD) in mice [50]. Besides inhibiting pro-inflammatory cytokines, ergosterol restored the activities of superoxide dismutase and reduced the content of malondialdehyde in serum and in the lung. Another study of ergosterol’s protective effect against the cigarette smoke extract-induced COPD suggested that protective effects may be related to the NF-κB/p65 signaling pathway [51].

The transcription factor Nrf2 plays an important role in controlling the expression of antioxidant genes, which ultimately leads to anti-inflammatory effects. Activation of the Nrf2 signaling pathway by ergosterol was shown to enhance cardiomyocyte resistance to oxidative stress in lipopolysaccharide- or isoproterenol-induced myocardial injury [52,53]. Oral administration of ergosterol (25 mg/kg/day) to mice for two weeks effectively delayed the progression of osteoarthritis through a mechanism involving activation of the Nrf2 pathway in primary chondrocytes [54].

Diabetic nephropathy is a chronic loss of kidney function in patients with diabetes mellitus. Ergosterol has been shown to attenuate kidney damage in diabetic mice [55,56]. It restored blood glucose and serum insulin levels and improved most biochemical and renal functional parameters. Xiong et al. [57] considered ergosterol as a potential hypoglycemic agent for the treatment of type 2 diabetes mellitus based on the discovery that it could promote glucose transporter type 4 translocation and expression, as well as glucose uptake via the PI3K (phosphatidylinositol 3-kinase) and Akt (protein kinase B) pathways. Hyperglycemia promotes the formation of advanced glycation end products (AGE) by crosslinking proteins and carbohydrates. Ergosterol prevented the suppression of oxidative stress in HSC-T6 cells and prevented age-related diseases such as liver fibrosis and diabetes [58].

An inhibitory effect of ergosterol against human recombinant aromatase (IC_50_ 8.1 μM) was observed in aromatase inhibitory assay [59]. Potential beneficial effects against ethanol hepatotoxicity were predicted by density functional theory calculations based on the ability of ergosterol to scavenge the •CH(OH)CH_3_ radical [60]. 

The following pharmacokinetic parameters were measured after a single oral administration (100 mg/kg) of ergosterol to rats: the area under the plasma concentration versus time curve from time 0 h to 36 h (AUC_0–36_) was 22.3 μg h mL^−1^, peak plasma concentration (C_max_) was 2.27 μg/mL, the elimination half-life (t_1/2_) was 5.90 h, and time to C_max_ (T_max_) was 8.00 h [61].

Ergosterol is an easily crystallized compound with low water and oil solubility. To increase its bioavailability, nano-sized delivery vehicles were suggested to overcome this limitation. Poly(lactide-co-glycolide) nanoparticle encapsulation allowed a 4.9-fold increase of oral bioavailability compared to free ergosterol [62]. The relative oral bioavailability of ergosterol-loaded nanostructured lipid carriers prepared using glyceryl monostearate and decanoyl/octanoyl glycerides by hot emulsification-ultrasonication was 277% higher than that of ergosterol itself [63]. 

In addition to being used as an active ingredient, ergosterol has also been tested as part of other drug delivery systems. The study of cellular uptake and in vitro cytotoxicity of cyclic arginine-glycine-aspartic and octa-arginine peptide-modified ergosterol-combined cisplatin liposomes showed their stability in serum and the strongest anti-lung cancer effect [39]. The encapsulation of chlorin e6 in self-assembled ergosterol nanoparticles resulted in a novel supramolecularly assembled photosensitizer [64]. When applied to cancer cells 4T1 and MCF-7, it showed remarkable in vitro phototoxicity with cell inhibition of about 73% and 92%, respectively. Evident in vitro antiproliferative activity was demonstrated for a mixture of sterols (consisting mainly of ergosterol and 22,23-dihydroergosterol) from popular edible mushroom *Flammulina velutipes* [65]. Encapsulation of the mixture increased the relative bioavailability of ergosterol and 22,23-dihydroergosterol to 163 and 244%, respectively.

Another way to increase the bioavailability of ergosterol is the preparation of its derivatives. Direct esterification of ergosterol and lauric acid led to the coupling product ergosterol laurate (**3a**) (Figure 2) with solubility in vegetable oil above 5.7 g/100 mL, while for ergosterol it was below 0.9 g/100 mL [66]. Esters of unsaturated fatty acids, ergosterol oleate (**3b**), ergosterol linoleate, and ergosterol linolenate were prepared by transesterification reaction using *Proteus vulgaris* K80 lipase [67]. Their solubility in the tricaprylin solvent was 11–16 times higher than that of the initial sterol. Another ergosterol ester, α-linolenic acid derivative, was prepared using *Candida* sp. 99-125 lipase as a biocatalyst [68].

The glucopyranosyl derivative **4** showed slightly higher activity in inhibiting LPS-induced NO production than ergosterol (**1**) (IC_50_ 16.6 and 14.3 μM, respectively) [69]. On the other hand, COX-1 enzyme inhibitory activity of **4** was weaker compared with that of the aglycone **1** [70].

Ergosterol adduct, ferulate **5**, was studied for the HMG-CoA reductase inhibitory activity, which was 1.93 times higher than that of oryzanol [71]. Another adduct **6**, derived from 2-naphthoic acid and ergosterol, showed stronger anti-tumor [72] and antidepressant [73] activities in vivo compared to ergosterol. 

The antiproliferative effects of some ergosterol dimers have been studied against the HT29 and MCF-7 cancer cell lines [74]. The most effective was dimer **7** for the HT29 cancer cell line with an IC_50_ value of 160 μM. Unfortunately, the results of comparing the activity with ergosterol itself were not reported.

### 2.2. Other Fungal Sterols

Sterol fraction of fungi is typically a mixture of sterols [75]. As a rule, ergosterol has been considered to be its dominant component. However, this is not true in all cases. There are at least four other taxon-specific sterols (cholesterol, 24-methylenecholesterol, 24-ethylcholesterol, and brassicasterol), which may be the main sterols in some fungal species [76]. Research on the biological or pharmaceutical uses of ergostane sterols has received much less attention compared to ergosterol or functionalized ergostanes. Only sterols that have attracted attention as objects for the further in-depth study will be considered here.

5,6-Dihydroergosterol or stellasterol (**8**) (Figure 3) is widely found as a minor ergostane constituent of many fungi, including sclerotia of *Polyporus umbellatus* [77], mycelium of *Cordyceps jiangxiensis* [78], *Stereum insigne* [79], *Eurotium rubrum* [80], fruiting bodies of *Stropharia rugosoannulata* [81], *Amauroderma amoiensis* [82], *Amauroderma subresinosum* [83], *Lasiosphaera fenzlii* [84], *Cortinarius xiphidipus* [85], *Pleurotus eryngii* [59], *Trametes versicolor* [86]. For practical purposes, a more suitable source of stellasterol (**8**) is its chemical synthesis from ergosterol [69,87]. 

Andrade et al. studied the effect of the purified *Marthasterias glacialis* extract and stellasterol (**8**) as its sterol constituent on inflammation in LPS-treated RAW 264.7 cells [88] and against human breast cancer (MCF-7) and human neuroblastoma (SH-SY5Y) cell lines [89]. The maximum anti-inflammatory effect was achieved when used in combination with unsaturated fatty acids [88]. In experiments with cancer cells, treatment with the extract markedly affected their growth, with stellasterol being responsible for the cell cycle arrest [89]. Yang et al. reported decreased NO production in LPS-treated RAW 264.7 cells with IC_50_ value of 15.1 μM and inhibition of iNOS and COX-2 [90].

The oxygen radical antioxidant capacity (ORAC) assay of components of the edible mushroom *Meripilus giganteus* revealed the highest antioxidant activity (4.94 mmol TE/g) for stellasterol (**8**) [91]. 

The study of the mechanism of anti-diabetic activity of the cosmopolitan woody polypore fungus *Ganoderma austral* showed that this may be due to its major component, stellasterol [92]. Its IC_50_ as an α-glucosidase inhibitor (315 μM) was close to that of acarbose (208 μM), which is an anti-diabetic drug used to treat diabetes mellitus. 

Stellasterol was also isolated from fruiting bodies of *Ganoderma lucidum* as pentadecanoate ester (**9**), which at a dose 100 mg/kg bw demonstrated moderate anti-inflammatory activity (60% inhibition) in carrageenan-induced paw edema [93]. 

Kim et al. conducted an extensive study of the effects of synthetically obtained stellasterol glucoside (**10**) and its analogs on skin inflammation [69,94,95,96]. It has been shown that **10** exhibits strong inhibitory activity against the production of nitric oxide (NO), which is a molecular mediator involved in inflammation. In addition, glucoside **10** suppressed the production of Th2-type chemokines CCL17 and CCL22. It was not cytotoxic up to a concentration of 100 μM, which makes it possible to consider **10** as a potential therapeutic agent for atopic dermatitis. Further studies in this area led to the discovery of galactosyl Δ^8(14)^-ergostenol (**11**) as the best candidate for the treatment of arthritis [97].

Ergostatrienol **12** (also named as antrosterol or EK100) is a quite common steroid in fungal sources. In particular, it was isolated from *Antrodia camphorate* [98,99,100], *Coprinus setulosus* [101], *Cordyceps militaris* [102], *Ganoderma resinaceum* [103], *Nigrospora sphaerica* [104], *Xylaria nigripes* [105]. 

Shih et al. showed that antrosterol (**12**) may be useful in the treatment of type 2 diabetes associated with hyperlipidemia [98]. Its use has led to a decrease in blood glucose and total cholesterol and triglyceride levels, an increase in the GLUT4 protein in skeletal muscle, and an improvement in insulin resistance. 

The anti-inflammatory properties of *Antrodia camphorata* mycelium, used in traditional Chinese medicine, are at least partially determined by the presence of antrosterol as one of its constituents. Similar to the action of corticosteroids, compound **12** reduced the expression of IL-6 and IL-1β in macrophages [106]. The mechanism of anti-inflammatory effect of **12** has also been studied by Kuo et al. [107]. Authors explained the observed effect by an increase in the activity of antioxidant enzymes such as catalase, superoxide dismutase, and glutathione peroxidase in liver tissue, and the reduction of the expression of iNOS and cyclooxygenase-2. The studies [108,109] also noted a decrease in the expression of the inflammatory factor NF-κB and inflammatory cytokines IL-6 and TNF-α. The mechanism of anti-inflammatory action of **12** was also investigated in LPS-stimulated RAW264.7 cells and *Drosophila* [102].

In experimental acute ischemic stroke model, antrosterol (**12**) reduced ischemic brain damage by decreasing the expression of p65NF-κB and caspase 3 and promoted neurogenesis and neuroprotection by activating PI3k/Akt-associated inhibition of GSK3 and activation of β-catenin [110]. Compound **12** was proposed as a potential therapeutic agent in intracerebral hemorrhage [111]. It had an inhibitory effect on the activation of the microglial c-Jun N-terminal kinase and attenuated the expression of brain cyclooxygenase, activation of matrix metalloproteinase and brain injuries in a model of intracerebral hemorrhage in mice. Long-term daily administration of **12** was shown to be safe and can be used as a potential ergogenic aid [112].

Hu et al. showed a strong cytotoxic effect of **12** against human U2OS lung osteosarcoma cells with IC_50_ value of 0.93 μM [105]. 

Cholesterol is a vital component of eukaryotic cells and its trafficking is an important issue for their proper functioning. 9-Dehydroergosterol (**13**) has proven to be a very convenient biochemical tool for studying cholesterol transport in living cells [113,114,115]. First of all, this is due to its own fluorescence because no additional moieties covalently attached to cholesterol are required. Second, 9-dehydroergosterol (**13**) mimics cholesterol very well, which is a consequence of its ability to stand upright in the membrane, almost identical to cholesterol.

Ano et al. found that extracts of dairy products fermented with *Penicillum candidum* have potent anti-inflammatory effect on microglia [116]. Repeated purification of the extracts led to the isolation of 9-dehydroergosterol (**13**) as an active principle responsible for the observed effect. Compound **13** significantly inhibited neurotoxicity and neuronal cell death induced by over-activated microglia, making it a valuable agent for the prevention of dementia. 

Dendritic cells play a key role in regulating the balance between tolerance and immune response. It has been shown that 14-dehydroergosterol (**14**) induces the transformation of dendritic cells in the bone marrow of mice and differentiates them into a tolerogenic type [117]. It can be helpful in preventing chronic inflammatory and autoimmune diseases.

She et al. isolated from the mangrove-derived fungus *Aspergillus* sp. two steroids having a 6/6/6/6/5 pentacyclic steroidal system [118]. Ergosterdiacid A (**15**) was supposed to be a natural Diels-Alder product derived from fumaric acid and ergostatetraene **14**. In vitro experiments showed that adduct **15** was active against *Mycobacterium tuberculosis* tyrosine phosphatase B (IC_50_ 15.1 μM) and had a strong anti-inflammatory effect by suppressing NO production at 4.5 μM.

A number of hybrids of 9-dehydroergosterol with polyketides have been isolated from natural sources. Two anthraquinone derivatives, evantrasterol A and B (**16** and **17**) (Figure 4), have been found in the endophytic fungus *Emericella variecolor* [119].

Elsebai et al. isolated nitrogenous metabolites of phenalenone, conio-azasterol (**18**), and S-dehydroazasirosterol (**19**), from the marine endophytic fungus *Coniothyrium cereal* [120]. Another nitrogenous hybrid of 9-dehydroergosterol fused through the morpholine ring with alternariol, pestauvicomorpholine A (**20**), was isolated from the fermentation product of the fungus *Pestalotiopsis uvicola* [121]. No cytotoxicity was detected for any of the tested compounds **16**–**20**.

## 3. Endoperoxides

Compounds containing a peroxide group are quite widespread among various natural substances, and steroids are not an exception [27]. Two 5α,8α-endoperoxides, ergosterol peroxide (EP, **21a**) and 9,11-dehydroergosterol peroxide (DHEP, **22a**) (Figure 5), are the most typical representatives of fungal steroids. Publications up to 2016 on the biological activity of EP (**5a**) have been thoroughly reviewed by Merdivan and Lindequist [24], and only the more recent literature regarding this compound will be discussed here. 

Biological studies of endoperoxides **21a** and **22a** have been aimed primarily at assessing their cytotoxic potential. Both compounds revealed quite high level of cytotoxicity in a wide range of cancer cells (Table 1). It should be noted that measurements of cell toxicity often vary significantly from laboratory to laboratory. Thus, for EP and cell line MCF-7 the values of IC_50_ varied from IC_50_ 1.18 μM [122] to 151 μM [123].

Attempts have been made to understand the cytotoxicity mechanism for **21a**, and some authors have concluded that more than one mechanism is at work. Obviously, the peroxide bridge plays a crucial role, bearing in mind that ergosterol is not cytotoxic. It was assumed that induction of apoptosis is the main cause of cytotoxicity [24]. Homolytic cleavage of the peroxide moiety in a reducing medium leads to the formation of reactive oxygen species (ROS), which are powerful internal stimuli for apoptosis. This has been confirmed, in particular, in experiments with MCF-7 cells [124]. Their treatment with **21a** at concentrations of 40–80 μg/mL led to an increase in the production of ROS in a dose-dependent manner and to the induction of apoptosis. The inhibitory properties of **21a** against A549 lung cancer cells were mediated by mitochondria-dependent apoptosis and autophagy [125]. EP also reduced LPS/ATP-induced proliferation and migration of A549 cells. A synergistic effect was observed when using EP with kinase inhibitor Sorafenib.

Based on ID_50_ values for the MCF-7 cell line (1.18 μM) compared to the MDA-MB-231 cell line (12.82 μM), EP (**21a**) was hypothesized to target estrogen receptors [122]. Its possible role as an ERα antagonist was suggested by Kim et al. based on the suppression of the increase in the viability of MCF-7 cells caused by 17β-estradiol [126].

Ergosterol peroxide (**21a**) and 9,11-dehydroergosterol peroxide (**22a**) were often isolated from the same fungal material, and on the whole both compounds exhibit similar biological properties. DHEP (**22a**) was slightly more cytotoxic than EP (**21a**) on the Hep 3B cell viability (IC_50_ 16.7 and 19.4 μg/mL, respectively) [127]. In experiments with BV-2 microglia cells, compound **22a** did not damage cell viability, although EP was cytotoxic to these cells [128]. Kobori et al. showed that **22a** selectively inhibits the growth of HT29 human colon adenocarcinoma cells without affecting normal human WI38 fibroblasts [129]. The inhibition was attributed to the induction of expression of an inhibitor of cyclin-dependent kinase 1A, thus causing cell cycle arrest and apoptosis. The rather strong cytotoxic effect of **22a** (IC_50_ 8.58 μM) on HeLa human cervical carcinoma cells was associated with the regulated expression of stathmin 1, a protein that is critical for the regulation of the cell cytoskeleton [130]. The mechanisms of **22a** cytotoxicity in A375 melanoma cells have been shown to be caspase-dependent and mediated via the mitochondrial pathway and include targeting of the induced differentiation protein of myeloid leukemia cells Mcl-1, release of cytochrome c, and activation of caspase-9 and -3 [131]. 

In experiments with a large number of cell lines EP possessed cytotoxic activity at the level of 1 μM and was more active in comparison with DHEP [132]. On the other hand, in the aromatase inhibitory assay 9(11)-double-bond enhances the inhibitory activity (IC_50_ > 100 μM vs. 32.6 μM for EP and DHEP, respectively) [59].

EP was thought to be one of the main compounds responsible for the antiproliferative effect of an ethanolic extract of the native New Zealand mushroom *Hericium novae-zealandiae* [133]. Two possible mechanisms of the observed effect have been proposed: apoptosis based on upregulation of CASP3, CASP8, CASP9, and anti-inflammation, as follows from downregulation of IL6 and upregulation of IL24.

Studying the cytotoxic effects on renal cell carcinoma cells, Zhang et al. found that EP treatment suppressed cell growth, colonization, migration and invasion, arrested the cell cycle, and triggered apoptosis [134]. This also means that several mechanisms can act for the same effect.

A similar situation with multiple pathways was observed in experiments with ovarian cancer cells [135]. Their treatment with **21a** inhibited nuclear β-catenin, thus decreasing the expression levels of cyclin D1 and c-Myc. Meanwhile, the level of protein tyrosine phosphatase SHP2 was increased in the treated cells, while the activity of Src kinase was suppressed. Thus, the antitumor effect of **21a** on ovarian cancer cells is due to both the β-catenin and STAT3 signaling pathways.

Significant inhibition of the formation of experimental lung metastases in vivo was found for EP (**21a**) [136]. The effect was attributed to inhibition of the NF-κB and STAT3 inflammatory pathways in 4T1 breast cancer cells.

EP was more effective than cisplatin in a mouse tumor model, inhibiting CT26 cell growth and improving the survival of tumor mice with no obvious side effects [137]. The growth of tumor cells of the gastrointestinal tract was suppressed due to the induction of apoptosis by the stress of the endoplasmic reticulum and mitochondria-dependent pathway.

**Table 1 molecules-27-02103-t001:** Cytotoxicity of fungal endoperoxides on different cell lines.

Compound	Cell Line	Origin *	Effect [Ref.]
**21a**	4T1	Mouse breast cancer	IC_50_ 9.06 μM [138]
	A549	Lung carcinoma	IC_50_ 17.04 μM [138], IC_50_ 17.2 μM [84], IC_50_ > 20 μM [139], IC_50_ 23 μM [125], IC_50_ 57 μM [140]
	B 16	Murine melanoma	IC_50_ 78.77 μM [141]
	B16F10	Murine melanoma	IC_50_ 55.8 μM [142]
	BGC823	Gastric cancer	IC_50_ 35.23 μg/mL [137]
	Eca-109	Esophageal carcinoma	IC_50_ 23.17 μg/mL [137]
	DU145	Prostate cancer	IC_50_ 21 μg/mL [133]
	HCT116	Colorectal carcinoma	IC_50_ 80.72 μM [142]
	HeLa	Cervical carcinoma	IC_50_ 13.6 μM [84], IC_50_ > 20 μM [139], IC_50_ 31 μM [125], IC_50_ > 50 μM [143], IC_50_ > 50 μM [138]
	Hep 3B	Hepatocellular carcinoma	IC_50_ 35.2 μg/mL [144]
	HepG2	Liver carcinoma	IC_50_ 13.19 μM [138], IC_50_ > 20 μM [139], IC_50_ 23.15 μM [145], IC_50_ 23.5 μM [146], IC_50_ 34 μM [147], IC_50_ 46.9 μM [144], IC_50_ 113 μM [123]
	HL-60	Promyelocytic leukemia	IC_50_ 39.4 μM [143]
	HT-29	Colon adenocarcinoma	IC_50_ 25.47 μM [137], IC_50_ > 50 μM [138]
	J5	Hepatocellular carcinoma	IC_50_ 33 μM [125]
	L1210	Mouse lymphotic leukemia	IC_50_ 36.40 μM [138]
	LNCap	Prostate cancer	IC_50_ 15 μg/mL [133], IC_50_ 35.53 μg/mL [141]
	LS180	Colon adenocarcinoma	IC_50_ 17.3 μg/mL [148]
	MDA-MB-231	Breast carcinoma	IC_50_ 12.82 μM [122], EC_50_ 18 μM [149], IC_50_ 24.75 μM [146], IC_50_ 44.6 μM [147]
	MCF-7	Breast cancer	IC_50_ 1.18 μM [122], IC_50_ 9.01 μM [138], IC_50_ 26 μM [140], IC_50_ 26.06 μM [145,146], IC_50_ 29 μM [125], IC_50_ 38.2 μM [143], IC_50_ 40 μM [124], IC_50_ 98.12 μM [142], IC_50_ > 100 μM [126,144], IC_50_ 151 μM [123]
	MGC-803	Gastric carcinoma	IC_50_ 15.2 μM [84]
	NCI 60 panel		significant activity against most tumor cell lines tested [132]
	PC3	Prostate cancer	IC_50_ 42 μg/mL [133]
	PC-3M	Prostatic carcinoma	IC_50_ 23.15 μM [123]
	RCC	Renal carcinoma	IC_50_ 30 μM [134]
	SK-Hep1	Liver cancer	IC_50_ 19.25 μM [145], IC_50_ 19.71 μM [146]
	SUM-149	Breast cancer	EC_50_ 9 μM [149], EC_50_ 20 μM [150]
	T-47D	Breast cancer	EC_50_ 19 μM [149]
**21b**	A549	Lung carcinoma	IC_50_ 14.21 μM [151]
	HCT-15	Colon adenocarcinoma	IC_50_ 17.49 μM [151]
	SK-MEL-2	Skin melanoma	IC_50_ 9.01 μM [151]
	SK-OV-3	Ovary malignant ascites	IC_50_ 15.11 μM [151]
	U87	Glioblastoma	20.1% inhibition at 100 μM [152]
**21c**	HepG2	Liver carcinoma	IC_50_ 12.34 (*n* = 1), 9.46 (*n* = 2), 6.74 (*n* = 3) μM [145]
	MCF-7	Breast cancer	IC_50_ 14.80 (*n* = 1), 13.70 (*n* = 2), 7.45 (*n* = 3) μM [145]
	SK-Hep1	Liver cancer	IC_50_ 10.43 (*n* = 1), 11.70 (*n* = 2), 5.92 (*n* = 3) μM [145]
**21d**	HepG2	Liver carcinoma	6.60 μM [145]
	MCF-7	Breast cancer	10.62 μM [145]
	SK-Hep1	Liver cancer	8.10 μM [145]
**21e**	MDA-MB-231	Breast carcinoma	EC_50_ 7 μM [149]
	SUM-149	Breast cancer	EC_50_ 2 μM [149]
	T-47D	Breast cancer	EC_50_ 16 μM [149]
**21f**	HCT-116	Colon carcinoma	IC_50_ 0.21 μM [153]
**21g**	SUM-149	Breast cancer	EC_50_ 12 μM [150]
**21h**	MDA-MB-231	Breast carcinoma	EC_50_ 10 μM [149]
	SUM-149	Breast cancer	EC_50_ 4 μM [149]
	T-47D	Breast cancer	EC_50_ > 10 μM [149]
**21i**	HepG2	Liver carcinoma	IC_50_ 0.85 μM [146]
	MCF-7	Breast cancer	IC_50_ 3.26 μM [146]
	MDA-MB-231	Breast carcinoma	IC_50_ 4.12 μM [146]
	SK-Hep1	Liver cancer	IC_50_ 1.75 μM [146]
**21j**	HepG2	Liver carcinoma	IC_50_ 2.83 μM [146]
	MCF-7	Breast cancer	IC_50_ 4.62 μM [146]
	MDA-MB-231	Breast carcinoma	IC_50_ 3.99 μM [146]
	SK-Hep1	Liver cancer	IC_50_ 0.92 μM [146]
**22a**	4T1	Mouse breast cancer	IC_50_ 9.31 μM [138]
	A375	Malignant melanoma	IC_50_ 9.46 μg/mL [131]
	A549	Lung carcinoma	IC_50_ 9.7 μM [84], IC_50_ 10.77 μM [138], IC_50_ 49 μM [125], IC_50_ 63 μM [140], IC_50_ 103.74 μM [154], IC_50_ 121.9 μM [155], No cytotoxicity [156]
	Calu-6	Lung carcinoma	IC_50_ 71.2 μM [155]
	Colo201	Colorectal adenocarcinoma	IC_50_ 13.02 μg/mL [131]
	H1264	Lung carcinoma	IC_50_ 92.3 μM [155]
	H1299	Lung carcinoma	IC_50_ 50.6 μM [155]
	HeLa	Cervical carcinoma	IC_50_ 7.6 μM [84], IC_50_ 8.58 μM [130], IC_50_ 35.82 μM [138], IC_50_ 37 μM [125]
	Hep 3B	Hepatocellular carcinoma	IC_50_ 16.7 μg/mL [127]
	HepG2	Liver carcinoma	IC_50_ 10.93 μM [138], IC_50_ 44.5 μM [147], IC_50_ 64.95 μM[154]
	HGC27	Gastric carcinoma	IC_50_ 26.47 μM [16]
	HT-29	Colon adenocarcinoma	IC_50_ 30.76 μM [138]
	J5	Hepatocellular carcinoma	IC_50_ 36 μM [125]
	L1210	Mouse lymphotic leukemia	IC_50_ 29.31 μM [138]
	MCF-7	Breast cancer	IC_50_ 3.3 μM [140], IC_50_ 8.40 μM [138], IC_50_ 16.89 μg/mL [131], IC_50_ 34 μM [125], IC_50_ 67.89 μg/mL [131], IC_50_ > 100 μM [126]
	MDA-MB-231	Breast carcinoma	IC_50_ 72.68 μM [154], IC_50_ 99 μM [16], IC_50_ 328 μM [147]
	MGC-803	Gastric carcinoma	IC_50_ 7.8 μM [84]
	Panc-28	Pancreatic adenocarcinoma	No cytotoxicity [156]
	SW620	Colorectal adenocarcinoma	IC_50_ 32.87 μg/mL [131]
**22b**	A549	Lung carcinoma	No cytotoxicity [156]
	A549	Lung carcinoma	IC_50_ 15.42 μM [151]
	HCT-15	Colon adenocarcinoma	IC_50_ 19.32 μM [151]
	Panc-28	Pancreatic adenocarcinoma	No cytotoxicity [156]
	SK-MEL-2	Skin melanoma	IC_50_ 12.96 μM [151]
	SK-OV-3	Ovary malignant ascites	IC_50_ 18.26 μM [151]
**27**	A549	Lung carcinoma	IC_50_ 5.26 μg/mL [12]
	MCF-7	Breast cancer	IC_50_ 5.15 μg/mL [12]
**28**	A549	Lung carcinoma	IC_50_ 0.26 μg/mL [157]
	HSC-3	Oral squamous cell carcinoma	IC_50_ 1.72 μg/mL [157]
	HSC-4	Oral squamous cell carcinoma	IC_50_ 1.94 μg/mL [157]
	MKN45	Stomach adenocarcinoma	IC_50_ 0.34 μg/mL [157]

* Human, if not stated otherwise.

Compound **21a** can be used as a radiosensitizer in the treatment of cervical cancer to reduce the toxic effects that occur after ionizing radiation therapy. Loss of viability of the cervical cell lines HeLa and CaSki was observed with increasing dose of **21a** [158].

Biological effects of EP (**21a**) and its Δ^9,11^-counterpart **22a** are not limited to their cytotoxic and anticancer properties. A detailed study on the bioactivity of the components of the truffle *Reddellomyces parvulosporus* revealed a number of EP activities, including anti-tyrosinase, anti-urease, anti-α-glucosidase, and anti-α-amylase ones [159]. Tyrosinase is an enzyme involved in the biosynthesis of melanin in humans, and its inhibitors are of interest for preventing excessive melanin production, as being active ingredients of skin whitening agents. Tyrosinase inhibitory activity (IC_50_: 202.37 μg/mL) of EP was also detected by Bai et al. [160].

Ng et al. reported the antidiabetic effect of **21a** that was due to the upregulation of glucose absorption and modulation of the PI3K/Akt, MAPK, and GLUT-4 signaling pathways [161].

EP was tested for its antileishmania activity against *Leishmania donovani* promastigotes and showed good activity with IC_50_ values of 9.43 μM [162]. The EP trypanocidal activity has been associated with its interaction with CYP51 [163]. The key structural moiety responsible for this is the peroxide bridge, which mediates interaction with the CYP51 heme binding site. At a later stage, this can cause the appearance of free radicals through homolytic cleavage at the O-O site, the pharmacophore responsible for the biological activity of **21a**.

Zhou et al. studied the immunoregulatory effect on inflammation caused by influenza A virus in human alveolar epithelial cells A549. EP (**21a**) was found to have anti-inflammatory effects and prevent virus-induced apoptosis by attenuating retinoic acid-inducible gene I signaling in infected cells [164].

Oral administration of EP to piglets infected with porcine delta-coronavirus resulted in a reduction in diarrhea, relief of intestinal damage, and a decrease in viral load in feces and tissues [165]. Wang et al. demonstrated that ergosterol peroxide prevents infection by suppressing porcine delta-coronavirus-induced autophagy via the p38 signaling pathway [166,167].

DHEP (**22a**) was found to exhibit strong anti-inflammatory effect in lipopolysaccharide-stimulated RAW264.7 cells [168,169,170]. It suppressed the production of NO even at 12.5 μM and pro-inflammatory cytokines interleukin 6 at 25 μM [168].

With age, mesenchymal stem cells in bone marrow tend to differentiate more into adipocytes than into osteocytes. Compounds **21a** and **22a** have been shown to inhibit the differentiation of mesenchymal stem cells toward adipocytes, which may be useful for the treatment of postmenopausal osteoporosis [171].

In experiments with 3T3-L1 mouse embryonic fibroblast cells, it was shown that EP inhibits triglyceride synthesis and reduces the accumulation of lipid droplets by suppressing adipogenesis [172].

The endoperoxides **21a** and **22a** were tested for their antibacterial activity [173,174,175,176,177]. The presence of a 9,11-double bond contributed to the increase in activity [173,177]. Thus, Δ^9,11^-derivative **22a** was more effective against *M. tuberculosis* H37Rv in comparison with **21a** (MIC 16 μg/mL and 64 μg/mL, respectively) [173]. Antitubercular activity of the fungus *Gliocladium* sp. MR41., was tested on M. tuberculosis. It was found to be due to EP (**21a**) with MIC 0.78 μg/mL [178].

Kim et al. isolated glucosides **21b** and **22b** from the Korean wild fungus *Xerula furfuracea* and tested their effects on adipogenesis and osteogenesis in a mouse mesenchymal stem cell line [10]. Both compounds were found to inhibit the differentiation of stem cells into adipocytes, which is of interest in the treatment of syndromes associated with menopause.

Significant antifungal and cytotoxic activities were reported for EP decanoate (**21f**) [153]. In disk diffusion test against *Candida albicans* culture, its MIC value was found to be 8.3 μg/disc that was comparable to clotrimazole (MIC 5.1 μg/disc). Compound **21f** showed also very good cytotoxicity against the HCT-116 cell line with IC_50_ value of 0.21 μM compared to doxorubicin (IC_50_ 0.06 μM).

In an attempt to improve antitumor activity, a number of derivatives of endoperoxide **21a** have been studied. Ergosterol peroxide sulfonamide **21g** was found to be more effective in reducing cancer cell viability than the parental endoperoxide **21a** [150]. Significantly, its toxicity to normal human BJ fibroblasts was minimal, indicating that **21g** targets cancer cells.

A series of EP analogs containing BODIPY or a biotin moiety was obtained by Rivas et al. as probes for cellular localization studies [149]. They demonstrated that EP is distributed across the cytosol with significant accumulation in the endoplasmic reticulum. In addition, the resulting compounds were tested for antiproliferative activity in breast cancer cell models. The most active ones were analogs **21e** and **21h** (Table 1).

Several adducts of EP with 7-*N*,*N*-diethylaminocoumarins have been obtained by Bu et al. [145]. Analogues **21c** and **21d** exhibited increased cytotoxicity compared to **21a**, which was explained by their localization mainly in mitochondria, as followed from fluorescence imaging. In addition, the piperazine derivative **21d** suppressed the formation, invasion, and migration of cell colonies, induced arrest of HepG2 cells in the G2/M phase, and increased the level of intracellular reactive oxygen species.

A number of EP 3-carbamate derivatives were obtained by Hu et al. [146]. They exhibited antiproliferative activity, which was 6–28 times stronger than that of the initial endoperoxide **21a** (Table 1). The most active compounds **21i** and **21j** contain piperazinyl and piperidinyl fragments.

A steroid-xanthone heterodimer, asperversin A (**23**), was isolated from the culture of *Aspergillus versicolor*, an endophytic fungus isolated from the marine brown alga *Sargassum thunbergii* [179]. Compound **23** was tested for biological activities against some bacterial and fungal strains with no noticeable effect.

Further structural modifications of steroidal molecule with retention of the 5α,8α-endoperoxide scaffold included changes in the carbon skeleton of the side chain [180,181]. Thus, 7-dehydrocholesterol peroxide, its acetate and hemisuccinate showed improved anticancer activity and selectivity over the corresponding derivatives of EP [180].

In comparison with the compounds **21a** and **22a**, 5α,9α-endoperoxides have been studied much less due to their lower availability. Compounds **24** and **25** (Figure 6) were isolated from the edible mushroom *Grifola gargal* and evaluated in the osteoclast-forming assay [182]. They inhibited osteoclast formation, which may be of interest for the prevention of osteoporosis. Endoperoxide **26**, isolated from the fruiting bodies of *Stropharia rugosoannulata*, protected neuronal cells by attenuating endoplasmic reticulum stress caused by thapsigargin, an inhibitor of the Ca^2+^-ATPase [81]. A significant cytotoxicity (Table 1) against A549 and MCF-7 cells was noted for the endoperoxide **27**, isolated from the fruiting body of a medicinal macro fungus *Ganoderma lingzhi* [12]. Agarol (**28**) was isolated as a tumoricidal substance from the mushroom *Agaricus blazei* [157]. Its cytotoxicity was evaluated against four cancer lines (Table 1). Agarol (**28**) was shown to induce apoptosis by increasing generation of ROS and release of apoptosis-inducing factor from the mitochondria to the cytosol.

## 4. Epoxides

The majority of compounds of this group are 5α,6α epoxides (Figure 7). Almost all of them contain a hydroxy- or keto group at C-7, Δ^8(9)^-, or Δ^8(14)^-double bond, and some 5α,6α-epoxides have a functionalized ring D. Other epoxides (4,5-, 5β,6β-, 8,9-, 8,14-, and 9,11-derivatives) are much less common in fungi (Figure 8). Compounds **29**–**59** were tested in various assays, including AChE inhibitory, cytotoxic, α-glucosidase inhibition, NO production inhibition, etc., (Table 2).

Bae et al. noted that the presence of an epoxy group in the tetracyclic skeleton of ergosterol derivatives increases their cytotoxic properties [183]. Isolation of a series of 5α,6α-epoxides from the macrofungus *Omphalia lapidescens* allowed to establish some structure activity relationship correlations [15]. The greatest cytotoxicity against a human gastric cancer cell line, HGC-27, was noted for the compound **30a** and **31a** containing an α-oriented hydroxyl group at C-7 and Δ^8(9)^- or Δ^8(14)^-double bond (Table 2). The transition to 7-ketones **33** and **36** led to a decrease in activity, and of both compounds, the derivative **33** without a double bond in the BC cycles was less active. The diepoxide **52** showed the least activity, which indicates the importance of the double bond for cytotoxic activity.

Epoxides **41**, **43a**, and **43b**, isolated from the culture of Basidiomycete *Polyporus ellisii*, were evaluated for cytotoxicity against five human cancer cell lines [184]. The first two compounds were practically inactive, while epoxide **41** exhibited strong activity against all tested cell lines with IC_50_ in the range from 1.5 to 3.9 μM (Table 2).

Ferreira et al. performed virtual screening experiments on low-molecular weight fungal constituents as potential MDM2 inhibitors [185]. The latter is an important negative regulator of the p53 tumor suppressor, and its inhibitors have significant anti-tumor activity. From the compounds studied, epoxide **29** returned one of the best docking scores.

Epoxide **31b** was found to exhibit potent inhibitory activity on the expression of mRNA of proprotein convertase subtilisin/kexin type 9 (PCSK9) [186]. The latter affects the low density lipoprotein receptor on the surface of liver cells, resulting in high level of low density lipoprotein cholesterol (LDL-C). PCSK9 inhibitors have been proposed as novel LDL-C-lowering agents for the treatment of hyperlipidemia. Compound **31b** showed activity with IC_50_ values of 8.23 μM, which was comparable with berberine (IC_50_ 8.04 μM) used as a positive control.

A number of epoxides were tested for their anti-inflammatory activity. As a rule, inhibition of TNF-α and NO production in LPS-stimulated RAW264.7 macrophage cells was used to evaluate anti-inflammatory effects. Epoxide **30c** showed superior inhibitory activity on NO production with IC_50_ value of 3.24 μM [103]. In the same experiment, the positive control L-NMMA, nitric oxide synthase inhibitor, revealed IC_50_ value of 49.86 μM. TNF-α secretion decreased after treatment of macrophage cells with epoxide **49**, which at 10 μM exhibited activity with inhibition value of 37.5% [187]. This was comparable to the positive control (52.5% at 1 μM) exerted by celecoxib, the cyclooxygenase-specific inhibitor.

**Table 2 molecules-27-02103-t002:** Sources and biological activity of fungal epoxides.

Compound	Fungal Source [Ref.]	Assays (Activity) [Ref.]
**29**	*Hericium erinaceus* [187,188]*, Chaetomium* sp. M453 [189], *Colletotrichum* sp. YMF432 [190], *Cordyceps sinensis* [191], *Stereum insigne* CGMCC5.57 [79]	**AChE inhibitory assay** (IC_50_ 67.8 μM) [190], **nematicidal and antibacterial assays** (no activity) [79]
**30a**	*Amauroderma subresinosum* [83], *Ganoderma lucidum* [147], *G. resinaceum* [103], *Grifola frondosa* [154]*, Omphalia lapidescens* [15], *Simplicillium* sp. YZ-11 [192]*, Stropharia rugosoannulata* [193], *Pleurotus eryngii* [6]	**α-glucosidase inhibition assay** (IC_50_ > 100 μM) [154], **cytotoxic assay** (HGC-27, IC_50_ 11.69 μM) [15], (MCF-7, IC_50_ 24.3 μM; NCI-H460, IC_50_ 19.8 μM; SF-268, IC_50_ 15.5 μM) [194], (A549, IC_50_ 35.99 μM; HepG2, IC_50_ 25.81 μM; MDA-MB-231, IC_50_ 29.73 μM) [154], (HepG2, IC_50_ 22.1 μM; MDA-MB-231, IC_50_ 20.3 μM) [147], **lettuce hypocotyl growth assay** (65–80% inhibition) [193], **NO production inhibition assay** (IC_50_ 12.4 μM) [6], (IC_50_ 19.77 μM) [103]
**30b**	*Ganoderma resinaceum* [103], *Stropharia rugosoannulata* [81]	**anti-fungal assay** (MIC 250 μM) [81], **NO production inhibition assay** (IC_50_ 17.23 μM) [103], **osteoclast-forming assay** [81]
**30c**	*Amauroderma amoiensis* [82], *Ganoderma resinaceum* [103]	**NO production inhibition assay** (IC_50_ 3.24 μM) [103]
**31a**	*Cortinarius glaucopus* [195], *Ganoderma lucidum* [147], *G. resinaceum* [103], *G. sinense* [196], *Grifola frondosa* [154]*, Hygrophorus russula* [183], *Leptographium qinlingensis* [197], *Omphalia lapidescens* [15]*, Simplicillium* sp. YZ-11 [192]*, Stropharia rugosoannulata* [193]*, Phellinus linteus* [198], *Pleurotus eryngii* [6], *Termitomyces microcarpus* [132]	**α-glucosidase inhibition assay** (IC_50_ > 100 μM) [154], **cytotoxic assay** (HGC-27, IC_50_ 18.97 μM) [15], (MCF-7, IC_50_ > 50 μM; NCI-H460, IC_50_ > 50 μM); SF-268, IC_50_ > 50 μM)-194], (A549, IC_50_ 69.11 μM; HepG2, IC_50_ 38.87 μM; MDA-MB-231, IC_50_ 46.76 μM) [154], (A549, IC_50_ 15.3 μg/mL; XF498, IC_50_ 15.1 μg/mL) [183], (HepG2, IC_50_ 50.6 μM; MDA-MB-231, IC_50_ 46.7 μM) [147], **HNE inhibitory assay** (IC_50_ 28.2 μM) [198], **lettuce hypocotyl growth assay** (61–78% inhibition) [193], **NCI 60 panel** [132], **NO production inhibition assay** (IC_50_ > 30 μM) [6], (IC_50_ 23.34 μM) [103], (IC_50_ > 40 μM) [196]
**31b**	*Ganoderma resinaceum* [103], *Hericium erinaceus* [187,188]*, Sparassis crispa* [186,199], *Phellinus linteus* [198], *Pleurotus eryngii* [6]	**cytotoxic assay** (MCF-7, IC_50_ > 50 μM) [194], (NCI-H460, IC_50_ > 50 μM) [194], (SF-268, IC_50_ > 50 μM) [194], **NO production inhibition assay** (IC_50_ 14.3 μM) [6], (IC_50_ 17.23 μM) [103], **PCSK9 mRNA expression** (inhibition, IC_50_ 8.23 μM) [186]
**31c**	*Hericium erinaceum* [200]	**PPAR transactivation assay** (EC_50_ 8.2 μM) [200]
**31d**	*Hericium erinaceum* [200]	**PPAR transactivation assay** (EC_50_ 6.4 μM) [200]
**32**	*Pleurotus eryngii* [59]	**aromatase inhibitory assay** (IC_50_ 17.3 μM) [59]
**33**	*Hericium erinaceum* [187], *Omphalia lapidescens* [15]	**cytotoxic assay** (HGC-27, IC_50_ 29.34 μM) [15], **HNE inhibitory assay** (IC_50_ 75.1 μM) [198], **TNF-α secretion assay** (inhibition value of 37.5% at 10 μM) [187]
**34**	*Grifola gargal* [182]	**osteoclast-forming assay** [182]
**35**	*Amauroderma subresinosum* [83]	**AChE inhibitory assay** (20.9% at 100 μM) [83]
**36**	*Omphalia lapidescens* [15]	**cytotoxic assay** (HGC-27, IC_50_ 23.41 μM) [15]
**37a**	*Pleurotus eryngii* [201]	**cytotoxic assay** (RAW264.7, IC_50_ > 30 μM) [201]
**37b**	*Stropharia rugosoannulata* [81]	**osteoclast-forming assay** [81]
**38**	*Grifola gargal* [182]	**cytotoxic assay** (HepG2, IC_50_ 200.9 μM; MDA-MB-231, IC_50_ 189.4 μM) [147], **osteoclast-forming assay** [182]
**39**	*Amauroderma subresinosum* [83], *Polyporus ellisii* [184]	**cytotoxic assay** (HL-60, IC_50_ 32.1 μM; SMMC-7721, A549, MCF-7, SW480, IC_50_ > 40 μM) [184]
**40**	*Pleurotus eryngii* [201]	**cytotoxic assay** (RAW264.7, IC_50_ > 30 μM) [201], **NO production inhibition assay** (IC_50_ 13.2 μM) [201]
**41**	*Polyporus ellisii* [184]	**cytotoxic assay** (HL-60, IC_50_ 1.5 μM; SMMC-7721, IC_50_ 3.9 μM; A549, IC_50_ 2.7 μM; MCF-7, IC_50_ 3.1 μM; SW480, IC_50_ 2.9 μM) [184]
**42**	*Phomopsis* sp. [202]	**α-glucosidase inhibition assay** (IC_50_ > 100 μM) [202]
**43a**	*Polyporus ellisii* [184], *Phomopsis* sp. [202]	**antibacterial assay** (MIC 28.2 μM against *Micrococcus tenuis*) [202], **cytotoxic assay** (HL-60, IC_50_ 32.1 μM; SMMC-7721, A549, MCF-7, SW480, IC_50_ > 40 μM) [184]
**43b**	*Ganoderma resinaceum* [103], *Polyporus ellisii* [184], *Phomopsis* sp. [202]	**cytotoxic assay** (HL-60, IC_50_ 18.8 μM; SMMC-7721, A549, MCF-7, SW480, IC_50_ > 40 μM) [184]
**44**	*Grifola gargal* [182]	**osteoclast-forming assay** [182]
**45**	*Pleurotus eryngii* [6]	**NO production inhibition assay** (IC_50_ > 30 μM) [6]
**46**	*Ganoderma lucidum* [147]	**cytotoxic assay** (HepG2, IC_50_ 138.3 μM; MDA-MB-231, IC_50_ 176.1 μM) [147]
**47**	*Amauroderma amoiensis* [82]	**AChE inhibitory assay** (14.63% inhibition at 100 μM) [82]
**48**	*Trametes versicolor* [168]	(NO inhibitory activity at 12.5 μM, IL-6 inhibitory effect at 25 μM) [168]
**49**	*Hericium erinaceus* [187,188]	**TNF-****α****secretion assay** (37.5% inhibition at 10 μM) [187]
**50**	*Hericium erinaceus* [187,188]*, Phellinus linteus* [198], *Stropharia rugosoannulata* [193]	**HNE inhibitory assay** (IC_50_ 35.2 μM) [198], **inhibition of lettuce hypocotyl growth** (no activity) [193]
**51**	*Ganoderma lucidum* [147], *Hericium erinaceum* [187]	**NO production inhibition assay** (moderate activity) [187]
**52**	*Aspergillus awamori* [203], *Omphalia lapidescens* [15]	**cytotoxic assay** (HGC-27, IC_50_ 58.43 μM) [15], (A549, IC_50_ 64 μM) [203]
**53**	*Hericium erinaceum* [187], *Pleurotus eryngii* [6]	**NO production inhibition assay** (IC_50_ > 30 μM) [6]
**54**	*Omphalia lapidescens* [15]	**cytotoxic assay** (HGC-27, IC_50_ 15.37 μM) [15]
**55**	*Pleurotus eryngii* [201]	**cytotoxic assay** (RAW264.7, IC_50_ > 30 μM) [201]
**56**	*Talaromyces stipitatus* [204]	**cytotoxic assay** (Hep3B, IC_50_ 4.75 μM; HepG2, IC_50_ 8.85 μM; Huh-7, IC_50_ 13.78 μM) [204]
**57**	*Aspergillus penicillioides* [205], *Ganoderma lingzhi* [12]	**antibacterial assay** (MIC 32 μg/mL against *Vibrio anguillarum*) [205], **cytotoxic assay** (A549, IC_50_ 8.57 μM; MCF-7, IC_50_ 6.09 μM) [12]
**58**	*Chaetomium* sp. [189]	**AChE inhibitory assay** (20–60% inhibition at 50 μg/mL) [189]
**59**	*Colletotrichum* sp. [206]	**AChE inhibitory assay** (18.2% inhibition at 100 μg/mL) [206]

Human neutrophil elastase (HNE) is a serine protease that can degrade extracellular matrix proteins such as collagen, fibronectin, etc. Inhibition of this enzyme can prevent the loss of skin elasticity, thereby preventing skin aging. Yoo et al. reported the HNE-inhibitory properties of *Phellinus linteus* mycelium components [198]. All three tested epoxides **31a**, **34,** and **50** showed significant activity with ID_50_ ranging from 28.2 to 75.1 μM.

Epoxides **30a**, **31a,** and **33** were isolated after anaerobic incubation of ergosterol peroxide (EP, **21a**) with rat intestinal flora [207]. Two of them (**30a** and **33**) were found to be more active against human colorectal cancer cells than the original EP. This means that EP’s strong anti-tumor properties may be (at least in part) due to its metabolic products.

A number of ergostane-type sterol fatty acid esters, including epoxides **31c** and **31d**, were isolated from the mushroom *Hericium erinaceum* and evaluated for their PPAR transactivational effects using a luciferase reporter system [200]. Oleyl and linoleyl esters **31c** and **31d** proved to be the most potent activators of the transcriptional activity of PPARs with EC_50_ values down to 6.4 μM.

## 5. Polyols

It should be kept in mind that the structures of ergostane-type steroids with hydroxyl and/or carbonyl group(s) given below do not fully reflect their diversity in fungal sources. A large number of compounds have been isolated before 2010; for a number of compounds isolated later, no data on biological activity are given, and for this reason they are not included in this review.

Many fungal ergostanes of this class are 5α-alcohols containing (an)other hydroxy (or a functionalized hydroxy) group(s) at C-6, C-9, and/or C-14 (Figure 9). 5α,6α Epoxides are their evident biosynthetic precursors. As a rule, rings A and B are trans-fused for most ergostanes of this group, with the exception of 5β-alcohols **77, 78, 84** (Figure 10). It should be noted that fomentarol B (**84**) has a cis-junction of ring B and C, which is rare among the ergostane type steroids [208].

Cerevisterol (**60**) is probably the best studied among 5α,6β-dihydroxy derivatives, as it is widespread in the fungal kingdom (Table 3). It should be noted that data on its cytotoxicity are inconsistent and sometimes contradictory. Thus, cerevisterol (**60**) showed significant activity with IC_50_ values of 1.1–1.9 μM against the BT-549, KB, SK-MEL, and SKOV-3 cancer cell lines [209]. On the other hand, it was practically inactive toward A549, HeLa, HepG2, and MCF-7 cells [210]. This inconsistence may be partly due to the diverse cell lines used by different authors. But a large difference was also observed in experiments with the same cell lines (e.g., reported IC_50_ values for HepG2 varied from 14.5 μM [211] to 174.6 μM [147]).

The results of studies of antimicrobial activity also vary quite a lot. Thus, in the course of searching for biologically active constituents of wood decaying mushrooms, *Trametes gibbosa* and *Trametes elegans*, Agyare et al. isolated cerevisterol (**60**) as a compound responsible for their antimicrobial activity [212]. It inhibited the growth of a number of bacteria with MICs ranging from 25 to 50 µg/mL (ciprofloxacin MICs were between 0.31 and 3.50 µg/mL). The sub-inhibitory concentration of **60** (3 µg/mL) modified the activity of commonly used antibiotics (either potentiating or reducing). Similar results with respect to antimicrobial activity of **60** were obtained by Zhou et al. [213]. On the other hand, no antimicrobial activity for cerevisterol (**60**) was reported in works [214,215].

To access the anti-inflammatory activity of cerevisterol (**60**), Lee et al. measured the levels of NO and PGE_2_ and the production of cytokines TNF-α, IL-1, and IL-6 in LPS-stimulated macrophages [216]. It was shown that **60** suppressed the LPS-induced production of NO and PGE2 and decreased the expression of pro-inflammatory cytokines.

**Table 3 molecules-27-02103-t003:** Sources and biological activity of fungal alcohols.

Compound	Fungal Source [Ref.]	Assays (Activity) [Ref.]
**60**	*Aspergillus fumigatus* [213], *A. versicolor* [179], *Cladosporium* sp. [217], *Clitocybe nebularis* [214], *Eurotium rubrum* [80], *Fomes fomentarius* [208], *Fusarium chlamydosporum* [209,218], *F. equiseti* [219], *F. solani* [216], *Ganoderma sinense* [196,220], *Glomerella* sp. [215], *Gomphus clavatus* [221], *Hericium erinaceum* [222,223], *Hypholoma lateritium* [224], *Lentinus polychrous* [225], *Leptographium qinlingensis* [197], *Leucocalocybe mongolica* [210], *Meripilus giganteus* [91], *Morchella esculenta* [226], *Omphalia lapidescens* [15], *Penicillium brasilianum* [227], *Pleurotus eryngii* [6], *P. tuber-regium* [228], *Polyporus umbellatus* [77,211], *Termitomyces microcarpus* [132], *Trametes gibbosa* and *T. elegans* [212], *Tricholoma populinum* [229], *Xylaria nigripes* [105]	**AChE inhibitory assay** (0.4% inhibition at 100 μg/mL) [80], **antibacterial assay** (no activity against *Streptococcus agalactiae*, *Staphylococcus epidermidis*, *Moraxella catarrhalis*, *Haemophilus influenzae*, and *Proteus mirabilis*) [214], (*S**. typhi*, *S**. aureus* and *A**. niger*, MICs 25 μg/mL each, *E**. faecalis*, MIC 50 μg/mL) [212], (*Bacillus subtilis* and *Escherichia coli*, MICs 64 μg/mL each; *Staphylococcus aureus*, MIC 32 μg/mL) [213], **cytotoxic assay** (A549, IC_50_ 94.75 μM; HeLa, IC_50_ 74.13 μM; HepG2, IC_50_ 46.58 μM; MCF-7, IC_50_ 63.76 μM) [210], (T47D, 50.2% inhibition at 30 μM) [229], (BT-549, 1.4 μM; KB, 1.90 μM; SK-MEL, 1.70 μM; SKOV-3, 1.1 μM) [209], (Caco-2, IC_50_ 37.56 μM; MCF-7, IC_50_ 32.4 μM; MDA-MB-231, IC_50_ 41.5 μM) [219], (HGC-27, IC_50_ 37.71 μM) [15], (MCF-7, IC_50_ 37.2 μM; PC-3, IC_50_ 80 μM) [221], (HepG2, IC_50_ 14.5 μM) [211], (HepG2, IC_50_ 174.6 μM; MDA-MB-231, IC_50_ 148.8 μM) [147], (SW1990, IC_50_ 32.81 μM; Vero, IC_50_ > 100 μM) [220], **NF-κB inhibitory assay** (IC_50_ 5.1 μM) [226], **HIV-inhibitory assay** (IC_50_ 9.3 μM) [230], **HNE inhibitory assay** (IC_50_ 77.5 μM) [198], **DPPH free radical-scavenging assay** (IC_50_ 11.38 μM) [222], **GIRK channel inhibitory assay** (13% inhibition at 10 μM) [224], **lipoxygenase inhibitory assay** (IC_50_ 5.46 μM) [218], **NO production inhibition assay** (IC_50_ > 40 μM) [196], (IC_50_ > 30 μM) [6], **ORAC assay** (antioxidant activity 1.94 mmol TE/g) [91], **PTP1B inhibitory activity assay** (IC_50_ 7.5 μg/mL) [77], **toxicity to *Pinus armandi* seedlings assay** (lethal rate 95% at 30 μg/mL) [197], **trap activity assay** (reduction to 28.1% from 332% in control cells) [223]
**61a**	*Aspergillus penicillioides* [205], *A. ustus* [231], *Aspergillus versicolor* [179], *Eurotium rubrum* [80], *Ganoderma lucidum* [232], *G. sinense* [233], *Hericium erinaceum* [223], *Omphalia lapidescens* [15], *Penicillium brasilianum* [227], *Pleurotus eryngii* [6], *Tricholoma populinum* [229], *Xylaria nigripes* [105]	**AChE inhibitory assay** (2.7% inhibition at 100 μg/mL) [80], **cytotoxic assay** (T47D, 23.7% inhibition at 30 μM; MDA-MB-231, 54.7% inhibition at 30 μM) [229], (U2OS, IC_50_ 6.0 μM) [105], (HGC-27, IC_50_ 4.17 μM) [15], [15], (HL-60, IC_50_ 22.4 μM; LLC, IC_50_ 55.3 μM; MCF-7, IC_50_ > 100 μM) [232], **HIV-inhibitory assay** (IC_50_ 3.8 μM) [230], **HNE inhibitory assay** (IC_50_ 14.6 μM) [198], **neuroprotective activity assay** (20.9% increase in cell viability against Aβ_25-35_-induced injury in SH-SY5Y neuroblastoma cells at the concentration 10 μM) [105], **NO production inhibition assay** (IC_50_ 20.4 μM) [6], (108.2% inhibitory rate at 10 μM) [230], **trap activity assay** (reduction to 74.8% from 332% in control cells) [223]
**61b**	*Fomes fomentarius* [208], *Omphalia lapidescens* [15]	**cytotoxic assay** (HGC-27, IC_50_ 25.50 μM) [15]
**61c**	*Eurotium rubrum* [80], *Hericium erinaceum* [223]	**AChE inhibitory assay** (17.9% inhibition at 100 μg/mL) [80], **trap activity assay** (reduction to 81.8% from 332% in control cells) [223]
**61d**	*Fusarium chlamydosporum* [218]	**lipoxygenase inhibitory assay** (IC_50_ 3.06 μM) [218]
**61e**	*Hericium erinaceum* [223]	**ORAC assay** (antioxidant activity 8.01 mmol TE/g at 10 μM) [223]
**62a**	*Eurotium rubrum* [80], *Fomes fomentarius* [208], *Hericium erinaceum* [223], *Hygrophorus russula* [183], *Omphalia lapidescens* [15]	**AChE inhibitory assay** (2.4% inhibition at 100 μg/mL) [80], **cytotoxic assay** (HGC-27, IC_50_ > 100 μM) [15], (HepG2, IC_50_ 196.9 μM; MDA-MB-231, IC_50_ 114.2 μM) [147], (A549, >30 μg/mL; XF498, >30 μg/mL) [183], **trap activity assay** (reduction to 138.9% from 332% in control cells) [223]
**62b**	*Hericium erinaceum* [200]	**PPAR transactivation assay** (EC_50_ 18.7 μM) [200]
**62c**	*Hericium erinaceum* [200]	**PPAR transactivation assay** (EC_50_ 20.6 μM) [200]
**63a**	*Ganoderma lucidum* [147], *Pleurotus eryngii* [6]	**cytotoxic assay** (HepG2, IC_50_ 62.5 μM; MDA-MB-231, IC_50_ 56.3 μM) [147], **NO production inhibition assay** (IC_50_ 29.8 μM) [6]
**63b**	*Ganoderma sinense* [220]	**cytotoxic assay** (SW1990, IC_50_ 5.05 μM; Vero, IC_50_ 22.59 μM) [220]
**64**	*Fomes fomentarius* [208], *Ganoderma lucidum* [147], *Hericium erinaceum* [187]	**cytotoxic assay** (HepG2, IC_50_ 156.4 μM; MDA-MB-231, IC_50_ 168.9 μM) [147], **TNF-α secretion assay** (33.7% inhibition at 10 μg/mL) [187]
**65**	*Clitocybe nebularis* [214], *Fomes fomentarius* [208], *Hericium erinaceum* [223], *Hygrophorus russula* [183], *Leptographium qinlingensis* [197], *Naematoloma fasciculare* [151], *Stropharia rugosoannulata* [81], *Tricholoma populinum* [229]	**antibacterial assay** (no activity against *Streptococcus agalactiae*, *Staphylococcus epidermidis*, *Haemophilus influenzae*, and *Proteus mirabilis*, marginal activity against *Moraxella catarrhalis*) [214], **anti-fungal assay** (MIC 500 μM) [81], **cytotoxic assay** (MCF-7, MDA-MB-231, T47D, no activity) [229], (HepG2, IC_50_ 129.7 μM; MDA-MB-231, IC_50_ 148.2 μM) [147], (A549, 17.1 μg/mL; XF498, 16.5 μg/mL) [183], (A549, 10.83 μM; HCT-15, 13.2 μM; SK-MEL-2, 10.39 μM; SK-OV-3, 12.16 μM;) [151]
**66**	*Ganoderma lucidum* [147]	**cytotoxic assay** (HepG2, IC_50_ 286.4 μM; MDA-MB-231, IC_50_ 216.5 μM) [147]
**67a**	*Omphalia lapidescens* [15]	**cytotoxic assay** (HGC-27, IC_50_ 12.71 μM) [15], (HepG2, IC_50_ 184.6 μM; MDA-MB-231, IC_50_ 224.2 μM) [147]
**67b**	*Hericium erinaceum* [200]	**PPAR transactivation assay** (EC_50_ 22.3 μM) [200]
**68a**	*Omphalia lapidescens* [15]	**cytotoxic assay** (HGC-27, IC_50_ 26.74 μM) [15]
**68b**	*Fomes fomentarius* [208]	**cytotoxic assay** (A549, IC_50_ 29.8 μM; MCF-7, IC_50_ 26.1 μM; NUGC-3, IC_50_ 24.1 μM) [208]
**69**	*Pleurotus eryngii* [6]	**NO production inhibition assay** (IC_50_ > 30 μM) [6]
**70**	*Hericium erinaceus* [187,188]	**TNF-α secretion assay** (25% inhibition at 10 μg/mL) [187]
**71**	*Penicillium granulatum* [234]	**cytotoxic assay** (no activity) [234]
**72**	*Hericium erinaceum* [187]	**TNF-α secretion assay** (36.7% inhibition at 10 μg/mL) [187]
**73**	*Coprinus setulosus* [101], *Ganoderma lipsiense* [235], *G. resinaceum* [103], *Xylaria nigripes* [105]	**antigiardial assay** (93.6% inhibition against *Giardia duodenalis* throphozoites) [235], **NO production inhibition assay** (IC_50_ 27.6 μM) [105], (IC_50_ 22.76 μM) [103], **tyrosinase inhibitory assay** (IC_50_ 6.9 μM) [236]
**74**	*Eurotium rubrum* [80]	**AChE inhibitory assay** (23.1% inhibition at 100 μg/mL) [80]
**75**	*Ganoderma resinaceum* [103]	**NO production inhibition assay** (IC_50_ 22.76 μM) [103]
**76**	*Penicillium granulatum* [234]	**cytotoxic assay** (no activity) [234]
**77**	*Omphalia lapidescens* [16]	**cytotoxic assay** (GES-1, IC_50_ > 50 μM; HGC-27, IC_50_ 12.28 μM; MDA-MB-231, IC_50_ 11.33 μM) [16]
**78**	*Omphalia lapidescens* [16], *Pleurotus eryngii* [6]	**cytotoxic assay** (GES-1, IC_50_ 28.0 μM; HGC-27, IC_50_ > 50 μM; MDA-MB-231, IC_50_ 24.85 μM) [16], **NO production inhibition assay** (IC_50_ > 30 μM) [6]
**79**	*Ganoderma duripora* [237], *Ganoderma lucidum* [232,238], *Phellinus linteus* [198]	**cytotoxic assay** (HL-60, IC_50_ 12.7 μM; LLC, IC_50_ 45.2 μM; MCF-7, IC_50_ > 100 μM) [232], (A549, MCF-7, PC-3, IC_50_ > 50 μM) [238], **HNE inhibitory assay** (IC_50_ > 100 μM) [198]
**80**	*Lasiodiplodia pseudotheobromae* [11]	**AChE inhibitory assay** (no activity) [11], **α-glucosidase inhibition assay** (no activity) [11]
**81**	*Penicillium granulatum* [234]	**cytotoxic assay** (A549, IC_50_ 5.5 μM) [234]
**82**	*Penicillium granulatum* [234]	**cytotoxic assay** (A549, BEL-7402, SHG-44, IC_50_ > 20 μM; ECA-109, IC_50_ 9.2 μM; HepG2, IC_50_ 7.0 μM) [234]
**83**	*Omphalia lapidescens* [16]	**cytotoxic assay** (GES-1, HGC-27, MDA-MB-231, IC_50_ > 50 μM) [16]
**84**	*Fomes fomentarius* [208], *Omphalia lapidescens* [16]	**cytotoxic assay** (MDA-MB-231, IC_50_ 140.86 μM) [16], **NO production inhibition assay** (98.77% inhibitory activity at 50 μM) [208]
**85**	*Penicillium chrysogenum* [239], *Penicillium granulatum* [240]	**anti-fungal assay** (8 mm diameter at 20 μg/disk) [239], **cytotoxic assay** (HeLa, IC_50_ 15 μg/mL; NCI-H460, IC_50_ 40 μg/mL; SW1990, IC_50_ 31 μg/mL) [239], (HepG2, IC_50_ 8.2 μM) [240]
**86**	*Penicillium granulatum* [234]	**cytotoxic assay** (no activity) [234]
**87**	*Penicillium granulatum* [234]	**cytotoxic assay** (A549, IC_50_ 8.0 μM; BEL-7402, IC_50_ 8.5 μM; ECA-109, IC_50_ 8.3 μM; HepG2, IC_50_ 6.7 μM; SHG-44, IC_50_ 4.8 μM) [234]
**88**	*Penicillium granulatum* [234]	**cytotoxic assay** (no activity) [234]

Yoo et al. studied the HNE-inhibitory potency of ergostanes isolated from the mycelium of *Phellinus linteus* [198]. Methyl ether **61a** revealed the highest activity among all tested compounds with an IC_50_ 14.6 μM, which was comparable with the positive control (epigallocatechin gallate, IC_50_ 12.5 μM). The corresponding alcohol **60** was five times less active than **61a**.

Kim et al. studied the inhibitory activity of steroids isolated from *Hericium erinaceum* against tartrate-resistant acid phosphatase (TRAP) [223]. The latter has become a promising target for the development of new therapeutics for the treatment of osteoporosis and other bone-related diseases. Compounds **60, 61a, 61c, 62a** at a concentration of 10 μM reduced TRAP activity in osteoclasts differentiated from RAW 264.7 cells, from 322% in control cells to 28–139% in treated cells.

Compared to 5α,6-diols, other fungal polyols (Figure 10) have been relatively less studied. As mentioned above, many ergostane steroids are found in both mushrooms and plants. In particular, this applies to triol **73** found in various fungal species [101,103,105,235]. Among sixty-three compounds isolated from bamboo *Sinocalamus affinis* and studied as inhibitors of estrogen biosynthesis, triol **73** showed the highest activity with an IC_50_ value of 0.5 μM [241]. It reduced the level of expression of aromatase mRNA in granulosa-like cells of human ovaries without affecting the catalytic activity of aromatase. This discovery makes the steroid **73** an interesting lead compound in the development of new agents for the treatment of estrogen-dependent cancers.

Studying the cytotoxicity of compounds isolated from the fruiting bodies of a medicinal mushroom *Ganoderma lucidum*, Min et al. selected the 2β,3α,9α-triol **79** for a more detailed evaluation [232]. Treatment with **79** in a dose-dependent manner inhibited the growth of HL-60 human premyelocytic leukemia cells with the IC_50_ value of 12.7 μg/mL. The effect was attributed to the induction of the apoptotic process, including activation of DNA fragmentation and caspase-3 activity.

## 6. Hydroxyketones

This group of ergostanes in the present review is divided into compounds containing two (Figure 11), three (Figure 12), and four or more (Figure 13) functional groups in the cyclic part of the steroid molecule. It should be borne in mind that such a classification is rather arbitrary and does not cover all the aspects that are relevant to these steroids.

The first 8β-hydroxyergosta-3-one type of steroid, cyathisterol (**89**), was isolated from the fruiting body of *Caluatia cyathiformis* [242]. Later, Ji et al. isolated from an algicolous strain of *Aspergillus ustus* a very similar but not identical compound called isocyathisterol (**90**) [231]. A detailed NMR study allowed to determine the configuration of all stereocenters in **90**. The authors concluded that the difference between the compounds **89** and **90** was in the C-9 and/or C-14 configuration.

Li et al. reported theoretical and experimental results on the properties of isocyathisterol (**90**) as inhibitor of isocitrate dehydrogenase IDH1 [233]. Mutations in this enzyme are associated with certain brain tumors, that makes IDH1 inhibitors as potential anticancer therapeutics for glioma patients. Based on the results of molecular virtual screening, isocyathisterol (**90**) had a low equilibrium dissociation constant of 18.40 μM, which confirmed the strongest binding to the IDH1 mutant. Kinetic studies showed that **90** inhibited the mutant enzyme in a noncompetitive manner.

Qi et al. isolated from spores of a medicinal mushroom *Ganoderma lucidum* a number of steroids possessing a 4,6,8(14),22-tetraene-3-one unit [243,244]. The obtained compounds called as ganodermasides A-D **91, 93, 110, 95** were tested for their antiaging effect on the yeast replicative lifespan assay (Table 4). All of them increased the average lifespan compared to negative control and exhibited effect similar to the known anti-aging substance, resveratrol.

A number of ergosterol metabolites including hydroxyketones **91, 93, 109** were isolated from a non-pathogenic filamentous fungus *Talaromyces stipitatus* [204]. Compounds **91, 93, 109** showed remarkable cytotoxic activities against hepatoma cell lines with IC_50_ values ranging down to 5.26 μM.

**Table 4 molecules-27-02103-t004:** Sources and biological activity of fungal hydroxyketones.

Compound	Fungal Source [Ref.]	Assays (Activity) [Ref.]
**89**	*Calvatia cyathiformis* [242]	
**90**	*Aspergillus ustus* [231], *Calvatia nipponica* [126], *Ganoderma sinense* [233], *Stereum hirsutum* [17], *Tricholoma imbricatum* [245]	**antibacterial assay** (against *E. coli*, *S. aureus*, and *A. salina* with inhibitory zones of 6.7, 5.7, and 5.1 mm, respectively, at 30 μg/disk) [231], **cytotoxic assay** (A549, IC_50_ 12.3 μM; HL-60, IC_50_ 18.7 μM; K562, IC_50_ 27.2 μM; MCF-7, IC_50_ 23.8 μM; SMMC-7721, IC_50_ 15.7 μM; SW480, IC_50_ 19.1 μM) [245], (MCF-7, IC_50_ > 100 μM) [126], (A549, IC_50_ 19.1 μM; HL-60, IC_50_ 14.6 μM; MCF-7, IC_50_ 20.4 μM; SMMC-7721, IC_50_ 19.0 μM; SW480, IC_50_ 25.7 μM) [17]
**91**	*Ganoderma lucidum* [243,244], *Talaromyces stipitatus* [204]	**cytotoxic assay** (Hep3B, IC_50_ 9.67 μM; HepG2, IC_50_ 11.83 μM) [204], **lifespan assay** (number of divisions of K6001 yeast strain cells before death: 8.2 in control, 8.9 at 1 μM, 11.4 at 10 μM, 9.4 at 100 μM) [244]
**92**	*Polyporus ellisii* [184]	**cytotoxic assay** (A549, HL-60, MCF-7, SMMC-7721, SW480, IC_50_ > 40 μM; HL-60, IC_50_ 22.8 μM) [184]
**93**	*Ganoderma lucidum* [243,244], *Talaromyces stipitatus* [204]	**cytotoxic assay** (Hep3B, IC_50_ 12.59 μM; HepG2, IC_50_ 18.95 μM; Huh-7, IC_50_ 32.81 μM) [204], **lifespan assay** (number of divisions of K6001 yeast strain cells before death: 8.2 in control, 9.1 at 1 μM, 11.1 at 10 μM, 9.6 at 100 μM) [244]
**94**	*Polyporus ellisii* [184]	**cytotoxic assay** (A549, HL-60, MCF-7, SMMC-7721, SW480, IC_50_ > 40 μM; HL-60, IC_50_ 17.8 μM) [184]
**95**	*Ganoderma lucidum* [243], *Phomopsis* sp. [246]	**antifungal assay** (MIC 64 μg/mL against *Fusarium avenaceum*, MIC 128 μg/mL against *Hormodendrum compactum*) [246], **lifespan assay** (number of divisions of K6001 yeast strain cells before death: 7.5 in control, 10.0 at 3 μM, 10.7 at 10 μM, 9.2 at 30 μM) [243]
**96**	*Chaetomium globosum* [247]	**cytotoxic assay** (A549, MG-63, SMMC-7721, IC_50_ > 50 μg/mL) [247]
**97**	*Colletotrichum* sp. [206], *Penicillium brasilianum* [227], *Pleurotus eryngii* [6], *Tricholoma imbricatum* [245]	**cytotoxic assay** (A549, IC_50_ 21.7 μM; HL-60, IC_50_ 7.9 μM) [245], **NO production inhibition assay** (IC_50_ 12.4 μM) [6]
**98**	*Tricholoma imbricatum* [245]	**cytotoxic assay** (HL-60, IC_50_ 25.7 μM; SMMC-7721, IC_50_ 27.3 μM; SW480, IC_50_ 37.7 μM) [245]
**99**	*Fomes fomentarius* [208], *Grifola frondosa* [48], *Phellinus linteus* [198]	**β-hexosaminidase release assay** (no activity) [48], **HNE inhibitory assay** (IC_50_ > 100 μM) [198], **NO production inhibition assay** (IC_50_ 32.87 μM) [208]
**100**	*Hericium erinaceum* [187]	**TNF-α secretion assay** (24.6% inhibition at 10 μg/mL) [187]
**101**	*Tricholoma imbricatum* [245]	**cytotoxic assay** (A549, IC_50_ 12.4 μM; HL-60, IC_50_ 12.2 μM; K562, IC_50_ 13.8 μM; MCF-7, IC_50_ 17.8 μM; SMMC-7721, IC_50_ 27.6 μM; SW480, IC_50_ 19.7 μM) [245]
**102**	*Chaetomium globosum* [247], *Phomopsis* sp. [202], *Tricholoma imbricatum* [245]	**α-glucosidase inhibition assay** (IC_50_ > 100 μM) [202], **cytotoxic assay** (A549, IC_50_ 20.72 μg/mL; MG-63, IC_50_ 15.34 μg/mL; SMMC-7721, IC_50_ 19.20 μg/mL) [247], (A549, IC_50_ 27.3 μM; HL-60, IC_50_ 23.6 μM) [245]
**103**	*Tricholoma imbricatum* [245]	**cytotoxic assay** (A549, IC_50_ 36.7 μM; HL-60, IC_50_ 16.6 μM; K562, IC_50_ 19.9 μM; MCF-7, IC_50_ 21.3 μM; SMMC-7721, IC_50_ 23.5 μM) [245]
**104**	*Pleurotus eryngii* [248]	**NO production inhibition assay** (weak activity) [248]
**105**	*Tricholoma imbricatum* [245]	**cytotoxic assay** (A549, IC_50_ 12.7 μM; HL-60, IC_50_ 7.7 μM) [245]
**106**	*Stereum hirsutum* [17]	**cytotoxic assay** (A549, IC_50_ 11.0 μM; HL-60, IC_50_ 3.1 μM; MCF-7, IC_50_ 12.3 μM; SMMC-7721, IC_50_ 9.0 μM; SW480, IC_50_ 13.4 μM) [17]
**107**	*Stereum hirsutum* [17]	**cytotoxic assay** (A549, HL-60, MCF-7, SMMC-7721, SW480, IC_50_ > 40 μM) [17]
**108**	*Gymnoascus reessii* [249], *Polyporus ellisii* [198], *Phomopsis* sp. [246]	**antifungal assay** (MIC 64 μg/mL against *Fusarium avenaceum*, MIC 256 μg/mL against *Aspergillus niger* and *Trichophyton gypseum*) [246], **antimalarial assay** (IC_50_ 3.4 μg/mL against *Plasmodium falciparum*) [249], **cytotoxic assay** (KB, IC_50_ 3.8 μM; MCF-7, IC_50_ 7.9 μM; NCI-H187, IC_50_ 1.9 μM; Vero, IC_50_ 3.3 μM) [249], **HNE inhibitory assay** (IC_50_ 20.5 μM) [198],
**109**	*Ganoderma resinaceum* [103], *Omphalia lapidescens* [15], *Talaromyces stipitatus* [204]	**cytotoxic assay** (Hep3B, IC_50_ 5.26 μM; HepG2, IC_50_ 6.29 μM; Huh-7, IC_50_ 16.23 μM) [204], (HGC-27, IC_50_ 16.93 μM) [15]
**110**	*Ganoderma lucidum* [243]	**lifespan assay** (number of divisions of K6001 yeast strain cells before death: 7.5 in control, 8.8 at 3 μM, 10.8 at 10 μM, 9.4 at 30 μM) [243]
**111**	*Colletotrichum* sp. [206], *Ganoderma sinense* [196], *Pleurotus eryngii* [250], *Psathyrella candolleana* [251], *Volvariella volvacea* [123]	**cytotoxic assay** (HepG2, IC_50_ 5.90 μM; SGC-7901, IC_50_ 12.03 μM) [123], (A549, HL-60, MCF-7, SMMC-7721, SW480, IC_50_ > 40 μM) [251], (RAW264.7, IC_50_ > 100 μM) [250], **NO production inhibition assay** (IC_50_ 28.5 μM) [196], (IC_50_ 100 μM) [250]
**112**	*Volvariella volvacea* [123]	**cytotoxic assay** (HepG2, IC_50_ 20.27 μM) [123]
**113**	*Ganoderma resinaceum* [103]	**NO production inhibition assay** (IC_50_ 35.19 μM) [103]
**114**	*Gliomastix* sp. [252]	**antiviral assay** (EV-71 virus, IC_50_ 17.8 μM) [252], **cytotoxic assay** (HL-60, IC_50_ 1.75 μM; DU-145, IC_50_ 7.37 μM; HeLa, IC_50_ 12.1 μM; MOLT-4, IC_50_ 6.53 μM) [252]
**115**	*Ganoderma philippii* [253]	**AChE inhibitory assay** (35.8% inhibition at 50 μg/mL) [253]
**116**	*Ganoderma resinaceum* [103]	**NO production inhibition assay** (IC_50_ 32.87 μM) [103]
**117**	*Pleurotus eryngii* [6]	**NO production inhibition assay** (IC_50_ 18.1 μM) [6]
**118**	*Penicillium purpurogenum* [254]	**cytotoxic assay** (A549, HepG2, MCF-7, IC_50_ > 100 μM) [254]
**119**	*Gymnoascus reessii* [249], *Phomopsis* sp. [246], *Talaromyces* sp. [255]	**antifungal assay** (MIC 128 μg/mL against *Candida albicans*, MIC 256 μg/mL against *Aspergillus niger* and *Hormodendrum compactum*) [246], **antimalarial assay** (IC_50_ 3.4 μg/mL against *Plasmodium falciparum*) [249], **cytotoxic assay** (KB, IC_50_ 20.4 μM; MCF-7, IC_50_ > 50 μM; NCI-H187, IC_50_ 12.5 μM; Vero, IC_50_ 19.3 μM) [249]
**120**	*Stereum hirsutum* [17], *Phomopsis* sp. [246]	**antifungal assay** (MIC 64 μg/mL against *Candida albicans* and *Hormodendrum compactum*, MIC 128 μg/mL against *Aspergillus niger*) [246], **cytotoxic assay** (A549, IC_50_ 27.8 μM; HL-60, IC_50_ 14.4 μM; MCF-7, IC_50_ > 40 μM; SMMC-7721, IC_50_ 32.0 μM; SW480, IC_50_ > 40 μM) [17]
**121**	*Lasiodiplodia pseudotheobromae* [11]	**AChE inhibitory assay** (no activity) [11], **α-glucosidase inhibition assay** (no activity) [11]
**122**	*Phomopsis* sp. [246]	**antifungal assay** (MIC 128 μg/mL against *Candida albicans* and *Fusarium avenaceum*, MIC 256 μg/mL against *Hormodendrum compactum*) [246]

## 7. Ketones

Most compounds of this group of ergostane-type steroids contain keto functions at C-3 and C-6, as well as a number of double bonds (Figure 14). Ergone (**124**) is probably the best studied among them [256]. It is found in many fungal sources (Table 5), usually with a content of less than 10 μg/g of mushroom fruit bodies. *Polyporus umbellatus*, in comparison with other mushrooms, contains the highest amount of this compound, which, under optimized conditions, can reach 86.9 μg/g [257]. For practical purposes, ergone (**124**) can be easily obtained through a three-step chemical synthesis from ergosterol [258]. Ergone has been reported to possess various activities (Table 5), including cytotoxic, anti-bacterial [205], anti-inflammatory [228,259], anti-malarial [249], diuretic [260] abilities, and protective effects of early renal injury [261,262].

Attempts were made to study the mechanism of its action. A strong anticancer effect of **124** to HepG2 cells was associated with the induction of G2/M cell cycle arrest and apoptosis in a caspase-dependent manner [270].

Wang et al. studied the effect of ergone (**124**) on lipopolysaccharide-induced acute lung injury [272]. Pretreatment of mice with **124** was found to reduce neutrophil recruitment, regulate the release of inflammatory cytokines, reduce pulmonary edema, and correct pulmonary insufficiency. The observed effects were associated with inhibition of the NLRP3 signaling pathway.

Ergone (**124**) was found to inhibit signaling pathways STAT3 and Src in head and neck cancer-initiating cells [263] that results in the reduction of their stemness properties and tumorigenicity and is of interest for the treatment of head and neck squamous cell carcinoma.

The variety of pharmacological activities prompted scientists to study pharmacokinetic properties of ergone. Fan et al. investigated the interactions between ergone and human serum albumin [273]. The latter is a carrier protein for many endogenous and exogenous molecules in blood and greatly affects the pharmacokinetics of drugs. Fluorescence spectroscopy revealed the binding of ergone to albumin, in which hydrogen bonds and hydrophobic interactions play a dominant role.

The following pharmacokinetic parameters were measured after a single oral administration (20 mg/kg) of ergone to rats: the area under the plasma concentration versus time curve from time 0 h to indefinite time (AUC_0–∞_) was 19.6 μg h mL^−1^, peak plasma concentration (C_max_) was 1.5 μg/mL, the elimination half-life (t_1/2_) was 5.90 h, and time to C_max_ (T_max_) was 3.8 h [266].

To improve the therapeutic effect of ergone, several drug delivery systems has been proposed [274,275]. The folate receptor is known to be overexpressed in a wide variety of cancers, which is the basis for the development of tumor-targeted drug delivery systems. One of them uses the most abundant protein in plasma, albumin. Folate-modified ergone bovine serum albumin nanoparticles showed increased cellular uptake, targeting ability and cytotoxicity toward KB cells [274]. An in vivo experiment showed a higher antitumor effect and less toxicity of ergone nanoparticles compared to free ergone. Another delivery system was based on the encapsulation of ergone in PEGylated liposomes [275]. Pharmacokinetic studies have shown that encapsulation provides a longer residence time of ergone in the blood, which leads to a more effective in vivo antitumor effect.

## 8. Fungal Steroids with a Transformed Side Chain

The metabolic transformations of the ergosterol side chain are not as diverse as those of the tetracyclic skeleton. As a rule, they include hydrogenation of the Δ^22^-double bond, its epoxidation, and hydroxylation of the terminal fragment (in most cases at C-25), as well as subsequent secondary transformations of the introduced functional groups.

Many steroids of this class of ergostanes are 25-hydroxy derivatives (Figure 15). Compounds **131**–**140** were tested in inflammatory, cytotoxic, and antibacterial assays, but showed no particular activity (Table 6).

The epoxide **143** (Figure 16) was isolated from a halotolerant fungus *Aspergillus flocculosus* PT05-1 cultured in a hypersaline medium [13]. It exhibited a moderate antibacterial and antifungal activity and a weak cytotoxicity against HL-60 and BEL-7402 cell lines.

An ochratoxin-ergosteroid heterodimer, ochrasperfloroid (**145**), was isolated from the sponge-derived fungus *Aspergillus flocculosus* 16D-1 [276]. It showed potent inhibitory effects on IL-6 production in LPS-induced cells and NO production in LPS-activated macrophages (Table 6). Fungi of *Aspergillus* genus have been the source of three more steroids with the same side chain, including asperfloroid (**146**) [277], asperflosterol (**148**) [278], and compound **147** [279]. Anti-inflammatory properties were identified for asperfloroid (**146**) and asperflosterol (**148**) (Table 6).

Three 18,22-cyclosterols, including aspersteroid B (**152**) and aspersteroid C (**153**), were isolated from the culture extract of *Aspergillus ustus* NRRL 275 [280]. Both compounds exhibited no cytotoxicity against MCF-7, HeLa, A549, and HT-29 cells. When analyzing the immunosuppressive effect on the proliferation of T- and B-lymphocytes in vitro, they showed activity from moderate to weak.

Two bis-epoxides, favolon (**149**) and favolon C (**150**), were isolated from the cultures of basidiomycete *Favolaschia calocera* BCC 36684 [281]. They were evaluated for a number of activities such as antimalarial, antitubercular, cytotoxic, but a positive result was obtained only in the antifungal assay.

A pair of steroidal epimers, penijanthoids A and B (**154** and **155**), were isolated from the marine-derived fungus *Penicillium janthinellum* [246]. Both compounds showed weak anti-*Vibrio* activity against three pathogenic *Vibrio* spp.

**Table 6 molecules-27-02103-t006:** Sources and biological activity of fungal steroids with a transformed side chain.

Compound	Fungal Source [Ref.]	Assays (Activity) [Ref.]
**131**	*Ganoderma sinense* [196]	**NO production inhibition assay** (IC_50_ 17.7 μM) [196]
**132**	*Ganoderma sinense* [196]	**NO production inhibition assay** (IC_50_ 32.4 μM) [196]
**133**	*Ganoderma sinense* [196]	**NO production inhibition assay** (IC_50_ 19.8 μM) [196]
**134**	*Fusarium chlamydosporum* [218]	**lipoxygenase inhibitory assay** (IC_50_ 7.23 μM) [218]
**136**	*Psathyrella candolleana* [251]	**cytotoxic assay** (A549, HL-60, MCF-7, SMMC-7721, SW480, IC_50_ > 40 μM) [251]
**136**	*Psathyrella candolleana* [251]	**cytotoxic assay** (A549, IC_50_ 23.4 μM; HL-60, IC_50_ 32.3 μM; MCF-7, IC_50_ 28.3 μM) [251]
**137**	*Psathyrella candolleana* [251]	**cytotoxic assay** (MCF-7, IC_50_ 22.3 μM; SMMC-7721, IC_50_ 29.3 μM) [251]
**138**	*Conocybe siliginea* [282]	**NO production inhibition assay** (IC_50_ > 40 μM) [282]
**139**	*Conocybe siliginea* [282]	**NO production inhibition assay** (IC_50_ > 40 μM) [282]
**140**	*Aspergillus alabamensis* [283]	**antimicrobial assay** (MIC 32 μg/mL against *Edwardsiella ictaluri*, MIC 64 μg/mL against *Vibrio alginolyticus*) [283]
**141**	*Mahonia fortune* [265]	**antibacterial assay** (MIC 100 μg/mL against *Staphylococcus aureus*) [265]
**142**	*Hymenoscyphus fraxineus* [284]	**antibacterial assay** (MIC 16.7 μg/mL against *Bacillus subtilis*, *Micrococcus luteus* and *Staphylococcus aureus*) [284], **cytotoxic assay** (L929, IC_50_ 24 μg/mL) [284]
**143**	*Aspergillus flocculosus* [13]	**antibacterial assay** (MIC 3.3 μg/mL against *Candida albicans*, 3.3 μg/mL against *Pseudomonas aeruginosa*, 1.6 μg/mL against *Enterobacter aerogenes*) [13]
**144**	*Trichoderma* sp. [230]	**HIV-inhibitory assay** (IC_50_ 41.6 μM) [230], **NO production inhibition assay** (10% inhibition at 10 μM) [230]
**145**	*Aspergillus flocculosus* [276]	**cytotoxic assay** (A549, IC_50_ 55.0 μM; HepG2, IC_50_ 23.6 μM) [276], **IL-6 immune-suppressive activity assay** (IC_50_ 2.02 μM) [276], **NO inhibitory activity assay** (IC_50_ 1.11 μM) [276]
**146**	*Aspergillus flocculosus* [277], *Chaetomium globosum* [285]	**cytotoxic assay** (A549, HepG2, THP-1, IC_50_ > 80 μM) [277], **IL-6 immune-suppressive activity assay** (IC_50_ 22 μM) [277]
**147**	*Aspergillus* sp. [279]	**antiviral assay** (no activity against H3N2 and EV71 viruses) [279]
**148**	*Aspergillus flocculosus* [278]	**cytotoxic assay** (A549, HepG2, THP-1, IC_50_ > 80 μM) [278], **IL-6 immune-suppressive activity assay** (IC_50_ 24 μM), **TNF-α secretion assay** (IC_50_ 28 μM) [278]
**149**	*Favolaschia calocera* [281]	**antifungal assay** (active in the agar diffusion test) [281]
**150**	*Favolaschia calocera* [281]	**antifungal assay** (active in the agar diffusion test) [281]
**151**	*Albatrellus confluens* [286]	**cytotoxic assay** (HL-60, PANC-1, A549, SK-BR-3, SMMC-7721, no activity) [286]
**152**	*Aspergillus ustus* [280]	**immunosuppressive assay** (ConA-induced T-cell proliferation, IC_50_ 22.49 μM; LPS-induced B-cell proliferation, IC_50_ 22.49 μM) [280]
**153**	*Aspergillus ustus* [280]	**immunosuppressive assay** (ConA-induced T-cell proliferation, IC_50_ 69.68 μM; LPS-induced B-cell proliferation, IC_50_ 69.68 μM) [280]
**154**	*Penicillium janthinellum* [246]	**antibacterial assay** (MICs 25.0–50.0 μM against three pathogenic *Vibrio* spp.) [246]
**155**	*Penicillium janthinellum* [246]	**antibacterial assay** (MICs 25.0–50.0 μM against three pathogenic *Vibrio* spp.) [246]
**156**	*Phoma* sp. [287]	**PTP inhibitory activity assay** (PTP1B, IC_50_ 25 μM each) [287]

## 9. Ergostanes with a Rearranged Tetracyclic Skeleton

Due to their intriguing structural complexity and promising biological activities, ergostanes with a rearranged tetracyclic carbon skeleton have become very attractive targets for chemists and biologists. A recent review [23] has covered this area quite thoroughly, but for consistency and completeness some results will be briefly discussed here.

Most ergostanes with a modified skeleton are highly functionalized compounds bearing three and more functional groups. A certain exception are aromatic 1(10→6)abeo-ergostane-type steroids **157**–**160** (Figure 17). Two of them, **157** and **158**, exhibited significant cytotoxicity toward murine colorectal CT26 and human leukemia K562 cancer cell lines (Table 7). Citreoanthrasteroid B (**158**) was also tested for the neuroprotective effects on PC12 cells injured by glutamate (15 mM) [288]. Compound **158** showed potential neuroprotective activities by inhibiting the death of injured PC12 cells with EC_50_ value of 24.2 μM.

Another 1(10→6)abeo-steroid, aspersteroid A (**161**), was isolated from the culture extract of *Aspergillus ustus* [280]. It exhibited moderate cytotoxicity on four cancer cell lines, antimicrobial activity against Gram-negative and Gram-positive bacteria and immunosuppressive activities against the proliferation of T and B lymphocyte cells in vitro (Table 7).

Three anthrasteroid glycosides, malsterosides A-C (**162a**–**c**), were isolated from the fungus *Malbranchea filamentosa* [289]. The sugar moiety in the side chain of all glycosides was found to be D-mannose and the glycoside **162c** contained N-acetyl-D-glucosamine at the C-3 position. Cytotoxicity studies were performed with the A549 and Hela cancer cell lines. A moderate cytotoxicity in both lines was noted for malsteroside A (**162a**).

Two 1(10→6)-abeo-14,15-secosteroids, asperfloketals A (**163**) and B (**164**), were found in the sponge-associated fungus *Aspergillus f locculosus* 16D-1 [290]. They exhibited no cytotoxicity against three tested cancer cell lines. Promising results were obtained in anti-inflammatory assays. Compounds **163** and **164** displayed stronger activity in the CuSO_4_-induced transgenic fluorescent zebrafish than ibuprofen used as a positive control.

A-nor-B-homo steroid **165** (Figure 18) containing a 10(5→4)-abeo-ergostane fragment was isolated from culture of basidiomycete *Polyporus ellisii* [184] and from the mangrove-derived fungus *Phomopsis* sp. MGF222 [202]. Compound **165** exhibited inhibitory activities against four out of five human cancer cell lines tested except A549 [184] (Table 7). It was also tested for the antibacterial activities against seven pathogenic bacteria and for the inhibitory activities against α-glucosidase, but no effect was observed [202].

Another A-nor steroid **166** was isolated from the fungus of *Lasiodiplodia pseudotheobromae* [11]. A distinguished structural feature of this compound is an additional δ-lactone ring between C-3 and C-9.

Two nearly identical steroids **167** and **168** featured a bicyclo[3.3.1]nonane motif were discovered in the fungi *Phomopsis* sp. TJ507A [7] and *Stereum hirsutum* [17]. The only difference in their structures is the presence of a methoxy group in phomopsterone A (**167**) instead of an ethoxy one in steresterone A (**168**). Compound **167** was tested for NO inhibitory activity. Steresterone A (**168**) was evaluated for the cytotoxicity against five human cancer cell lines. Both compounds showed no activity in the respective tests.

Three C25 steroids, neocyclocitrinols E-G (**169**–**171**) were isolated from endophytic fungus *Chaetomium* sp. M453 [189]. All compounds were tested for AChE inhibitory activities and cytotoxicity, however, no effect was found.

Cheng et al. isolated from *Ganoderma theaecolum* ganotheaecolin A (**173**), having a naphtho[1,8-ef]azulene ring system steroid [291]. At a concentration of 10 μM, it showed activity to promote neurite growth in PC12 cells, comparable to that of nerve growth factor used as control.

A new steroid sarocladione (**174**) bearing a 5,10:8,9-diseco moiety was isolated from the deep-sea-derived fungus *Sarocladium kiliense* [292]. The initially proposed configuration at C-3 and C-7 proved to be incorrect and was revised to 3*S*,7*R* through the chemical synthesis [293]. Cytotoxic studies of compound **174** revealed no apparent cellular toxicities.

Lin et al. isolated from the sponge-derived fungus *Aspergillus flocculosus* 16D-1 two 11(9→10)-*abeo*-5,10-secosteroids, aspersecosteroids A (**175**) and B (**176**) [278], a characteristic structural feature of which was the presence of a dioxatetraheterocyclic ring system. Both compounds were non-cytotoxic at the concentrations up to 40 μM and showed a strong inhibitory effect on the production of TNF-α and IL-6.

Spiroseoflosterol (**177**) (Figure 19), having a unique spiro[4.5]decan-6-one moiety, was isolated from the fruiting bodies of *Butyriboletus roseoflavus* [294]. It showed a strong cytotoxic effect on HepG2 cell line (IC_50_ 9.1 μM), which was comparable to that of sorafenib (IC_50_ 5.5 μM) used as a positive control. Moreover, spiroseoflosterol (**177**) was active against sorafenib-resistant Huh7/S cells with an IC_50_ value of 6.2 μM, that makes it a promising candidate for antihepatoma drug development.

Calvatianone (**178**), featuring a contracted tetrahydrofuran B-ring, was found in a rare mushroom *Calvatia nipponica* [126]. It showed a weak cytotoxicity against MCF-7 with IC_50_ > 100 μM (Table 7).

Another compound with a five-membered B ring, laschiatrion (**179**), was isolated from fermentations of *Favolaschia* sp. [281,295]. It was not active in antibacterial and cytotoxic assays, but exhibited antifungal activity in the agar diffusion test [281].

7-Nor-ergosterolide (**180**), featuring a pentalactone B-ring system, was found in the culture extract of an endophytic fungus *Aspergillus ochraceus* EN-31 [296] and a halotolerant fungus *Aspergillus flocculosus* PT05-1 [13]. Compound **180** showed pronounced cytotoxic and antibacterial properties.

A characteristic structural feature of erinarol J (**181**), isolated from the dried fruiting bodies of *Hericium erinaceum*, is the presence of 6,8-dioxabicyclo[3.2.1]oct-2-ene moiety [187]. Biotests have shown potent anti-inflammatory activity of **181** due to the inhibition of TNF-α secretion and NO production.

The first natural 5,6-secosteroid, eringiacetal A (**182**), was isolated from the fruiting bodies of mushroom *Pleurotus eryngii* [250]. Biological assays showed its modest cytotoxicity and ability to inhibit NO production.

Herbarulide (**183**) was first isolated from the endophytic fungus *Pleospora herbarum* as a compound having a campestane side chain [297]. Later the same structure was assigned to one of the constituents of the Taiwanese fungus *Antrodia camphorate* [298]. The correct structure of herbarulide (**183**) was proposed by Chen and Liu who isolated it from the fungus *Stereum hirsutum* [17]. The assignment was based rather on the assumption that the C-24 stereocenter of the starting ergosterol will remain unchanged during the transformations in the cyclic part. Finally, the correct structure of **183** was confirmed by its chemical synthesis [299]. Compound **184**, structurally very close to herbarulide (**183**), was isolated from the fruiting bodies of *Ganoderma resinaceum* [103].

Solanioic acid (**185**) is a degraded and rearranged steroid isolated from laboratory cultures of the fungus *Rhizoctonia solani* [300]. An important feature of its biological activity is antibacterial effect against methicillin-resistant *Staphylococcus aureus*. The latter is a cause of infection that is difficult to treat due to resistance to many antibiotics.

Tricholumin A (**186**) was isolated from the alga-endophytic fungus *Trichoderma asperellum* [301]. The only structural element of the parent ergosterol that remained after a number of metabolic stages of its biosynthesis is cycle A. The rest of the molecule, including a fragment of the side chain, has undergone deep transformations. Inhibitory properties of **186** against harmful microalgae and weak antibacterial activity against five aquatic pathogens were found.

Dankasterone A (**187**) (Figure 20) was first isolated from a fungal strain of *Gymnascella dankaliensis* derived from the sponge *Halichondria japonica* [302]. The initial erroneous assignment of stereochemistry at C-24 was corrected from *S* to *R* in a follow-up work by these authors [303]. Subsequently, compound **187** was repeatedly isolated from fungal sources as one of the ergostane constituents (Table 7). The only structural difference between **187** and dankasterone B (**188**) is the saturated ring A. From the endophytic fungus *Phomopsis* sp. TJ507A was also isolated phomopsterone B (**190**) differing from **187** by the presence of a methyl group at C-23 [7]. Dankasterone A (**187**) showed promising anticancer activities with IC_50_ down to 2.3 μM on a range of cancer cell lines (Table 7). Structure activity relationship studies of dankasterones A and B showed that the Δ^4^-double bond is essential for high cytotoxicity against the cancer cell lines tested. Carbonyl groups in dankasterone B (**188**) were other structural elements important for the high biological activity, because products of its NaBH_4_ reduction were not cytotoxic [17]. Phomopsterone B (**190**) was tested for inflammatory activity and showed promising results in iNOS inhibitory and NO production inhibition assays [7].

At first glance, the carbon skeleton of periconiastone A (**189**) [304] looks completely different from that of dankasterone B (**188**). In fact, compound **189** is available from **188** in one step via the intramolecular aldol reaction [305], which is also evidently realized in the course of its biosynthesis. So far, periconiastone A (**189**) has been tested for anti-inflammatory and antibacterial activities. Positive results were obtained in an antibacterial assay against Gram-positive bacteria [304].

An 8,14-seco-steroid, childinasterone A (**191**), was isolated from fruiting bodies of the ascomycete *Daldinia childiae* [306]. It showed no activity in cytotoxic studies and exhibited strong inhibition of NO production (IC_50_ value of 21.2 μM versus 41.5 μM for L-NMMA used as a positive control).

9,11-Secosteroids are quite common in sea sponges [22], but rather rare in fungal sources. The first such an ergostane **192** was isolated from king trumpet mushroom *Pleurotus eryngii* [6]. Compound **192** exhibited NO inhibitory activity similar to that of L-NMMA and revealed no cytotoxicity. Another 9,11-secoergostane (**193**), found in the fruiting bodies of *Pleurotus eryngii*, displayed similar profile of biological activity [6].

Three steroids with a rearranged ring B, eringiacetal B (**194**), matsutakone (**195**), and pleurocin B (**196**), were isolated from the fruiting bodies of *Pleurotus eryngii* by Tanaka et al. [248]. All three compounds revealed inhibitory activity on production of NO which was stronger than that of L-NMMA. The 13,14-seco-13,14-epoxysteroid, eringiacetal B (**194**), was most active with an IC_50_ of 13.0 μM compared to 23.9 μM for the L-NMMA positive control.

An 8(14→15)-abeo-steroid, asperflotone (**197**), was obtained from the solid culture of *Aspergillus flocculosus* 16D-1 [277]. Its characteristic structural feature is a rearranged bicyclo[4.2.1]non-2-ene ring system. Compound **197** was tested on three cancer cell lines with no cytotoxic effects. In immune-suppressive activity assay, asperflotone (**197**) exhibited inhibitory effects on IL-6 secretion.

The 15(14→22)abeo-steroid framework is common for ergostanes **198**–**203** (Figure 21), collectively referred to as strophasterols. It took some effort to establish the correct structures of these structurally related compounds. Strophasterols A–D (**198**–**201**) were first isolated from the mushroom *Stropharia rugosoannulata* [307]. The structure of strophasterin A (**198**) was established by X-ray crystallographic analysis. Comparison of the NMR data made it possible to assign the structure of **199** as the C-22 isomer of strophasterol A that was later confirmed by X-ray analysis [193]. Structure of strophasterol C (**200**) was proposed based on NOE correlations by Aung et al., who isolated it from the basidiomycete *Cortinarius glaucopus* together with glaucoposterol A (**203**) [195]. Additional evidence for the structure of **200** was obtained by its chemical synthesis [308]. Two more steroids with a strophastane skeleton, strophasterol E (**202**) and strophasterol F (**203**), were isolated from the fruiting bodies of *Pleurotus eryngii* [201]. Their structures were determined by X-ray analysis of the corresponding tris-p-bromobenzoate derivatives. Structural elucidation of strophasterol D (**201**) was done by comparing it with a synthetically prepared sample [309]. This work also showed that glaucoposterol A and strophasterol F are the same compound (**203**).

So far, the biological activity of strophasterols has been studied only marginally. Strophasterol A (**198**) showed a dose-dependent inhibitory effect on the toxicity of thapsigargin. The latter is known to disrupt the balance of the Ca^2+^ concentration in the endoplasmic reticulum that is especially harmful to neuronal cells. Under the action of strophasterol A (**198**), an increase in cell viability by 10.3% compared with the control was noted [307]. Strophasterols E and F were tested for anti-inflammatory activity, but showed no promising results [201].

A 15(14→11)-abeo-ergostane, penicillitone (**204**), was isolated from the culture of the fungus *Penicillium purpurogenum* SC0070 [254]. It was evaluated for cytotoxicity against three cancer lines and showed good potency with IC_50_ ranging from 4.44 to 5.98 μM. In addition, compound **204** was active in the inflammatory assay on the production of TNF-α and IL-6. At the concentration of 5 μM it reduced their secretion by 70.7% and 96.6%, respectively. For comparison, inhibition rates of the positive control dexamethasone at 100 μM were 87.3% and 96.7%, respectively. This makes promising further in-depth study of penicillitone (**204**) as an anti-inflammatory or antitumor agent.

**Table 7 molecules-27-02103-t007:** Sources and biological activity of fungal steroids with a rearranged tetracyclic carbon skeleton.

Compound	Fungal Source [Ref.]	Assays (Activity) [Ref.]
**157**	*Antrodia camphorata* [310], *Aspergillus ustus* [231], *Gibberella zeae* [311]	**cytotoxic assay** (CT26, IC_50_ 15.3 μM; K562, IC_50_ 19.9 μM) [310]
**158**	*Antrodia camphorata* [310], *Penicillium citreo-viride* [312], *Phyllosticta capitalensis* [288]	**cytotoxic assay** (CT26, IC_50_ 18.2 μM; K562, IC_50_ 12.5 μM) [310], **neuroprotective activity assay** (EC_50_ 24.2 μM) [288]
**159a**	*Aspergillus ustus* [231]	
**159b**	*Aspergillus ustus* [231]	
**160**	*Penicillium citreo-viride* [312]	
**161**	*Aspergillus ustus* [280]	**antimicrobial assay** (*Candida albicans*, MIC_50_ 17.24 μg/mL; *Escherichia coli*, MIC_50_ 17.24 μg/mL; *Staphylococcus aureus*, MIC_50_ 15.51 μg/mL) [280], **cytotoxic assay** (A549, IC_50_ 40.32 μM; Hela, IC_50_ 26.09 μM; HT-29, IC_50_ 43.58 μM; MCF-7, IC_50_ 32.03 μM) [280], **immunosuppressive assay** (ConA-induced T-cell proliferation, IC_50_ 23.61 μM; LPS-induced B-cell proliferation, IC_50_ 23.61 μM) [280]
**162a**	*Malbranchea filamentosa* [289]	**cytotoxic assay** (A549, IC_50_ 38.6 μM; Hela, IC_50_ 28.1 μM) [289]
**162b**	*Malbranchea filamentosa* [289]	**cytotoxic assay (A549, Hela, no activity) [289]**
**162c**	*Malbranchea filamentosa* [289]	**cytotoxic assay (Hela, IC50 76.9 μM) [289]**
**163**	*Aspergillus flocculosus* [290]	**anti-inflammatory assay [290]**
**164**	*Aspergillus flocculosus* [290]	**anti-inflammatory assay** [290]
**165**	*Phomopsis* sp. [202], *Polyporus ellisii* [184]	**α-glucosidase inhibition assay** (IC_50_ > 100 μM) [202], **cytotoxic assay** (A549, IC_50_ > 40 μM; HL-60, IC_50_ 17.1 μM; MCF-7, IC_50_ 23.3 μM; SMMC-7721, IC_50_ 21.3 μM; SW480, IC_50_ 16.3 μM) [184]
**166**	*Lasiodiplodia pseudotheobromae* [11]	
**167**	*Phomopsis* sp. [7]	**NO production inhibition assay** (IC_50_ > 25 μM) [7]
**168**	*Stereum hirsutum* [17]	**cytotoxic assay** (A549, HL-60, MCF-7, SMMC-7721, SW480, IC_50_ > 40 μM) [17]
**169**	*Chaetomium* sp. [189]	**cytotoxic assay** (A549, HL-60, MCF-7, SMMC-7721, SW480, IC_50_ > 40 μM) [189]
**170**	*Chaetomium* sp. [189]	**cytotoxic assay** (A549, HL-60, MCF-7, SMMC-7721, SW480, IC_50_ > 40 μM) [189]
**171**	*Chaetomium* sp. [189]	**cytotoxic assay** (A549, HL-60, MCF-7, SMMC-7721, SW480, IC_50_ > 40 μM) [189]
**172**	*Xylaria* sp. [313]	
**173**	*Ganoderma theaecolum* [291]	**neurite outgrowth-promoting assay in PC12 cells** (stimulated cell differentiation with a maximum effect at 10 μM) [291]
**174**	*Sarocladium kiliense* [292]	**cytotoxic assay** (Bel-7402, ECA-109, HeLa, PANC-1, SHG-44, no activity) [292]
**175**	*Aspergillus flocculosus* [278]	**cytotoxic assay** (A549, HepG2, THP-1, IC_50_ > 80 μM) [278], **IL-6 immune-suppressive activity assay** (IC_50_ 21 μM), **TNF-α secretion assay** (IC_50_ 28 μM) [278]
**176**	*Aspergillus flocculosus* [278]	**cytotoxic assay** (A549, HepG2, THP-1, IC_50_ > 80 μM) [278], **IL-6 immune-suppressive activity assay** (IC_50_ 26 μM), **TNF-α secretion assay** (IC_50_ 31 μM) [278]
**177**	*Butyriboletus roseoflavus* [294]	**cytotoxic assay** (HepG2, IC_50_ 9.1 μM; Huh7/S, IC_50_ 6.2 μM; L02, IC_50_ 22.8 μM) [294]
**178**	*Calvatia nipponica* [126]	**cytotoxic assay** (MCF-7, IC_50_ > 100 μM) [126]
**179**	*Favolaschia calocera* [281], *Favolaschia* sp. [295]	**antifungal assay** (activity against *Candida albicans*, *Cryptococcus neoformans*, etc. at concentrations of 10–50 μg/mL) [295]
**180**	*Aspergillus flocculosus* [13], *Aspergillus ochraceus* [296]	**antibacterial assay** (MIC 1.9 μg/mL against *Candida albicans*, 7.5 μg/mL against *Pseudomonas aeruginosa* and *Enterobacter aerogenes*) [13], **cytotoxic assay** (BEL-7402, IC_50_ 17.7 μM; HL-60, IC_50_ 12.4 μM) [13], (NCI-H460, IC_50_ 5.0 μg/mL; SMMC-7721, IC_50_ 7.0 μg/mL; SW1990, IC_50_ 28.0 μg/mL) [296]
**181**	*Hericium erinaceum* [187]	**NO production inhibition assay** (38.4% inhibition at 10 μg/mL) [187], **TNF-α secretion assay** (43.3% inhibition at 10 μg/mL) [187]
**182**	*Pleurotus eryngii* [250]	**cytotoxic assay** (RAW264.7, IC_50_ 25.6 μM) [250], **NO production inhibition assay** (IC_50_ 19.9 μM) [250]
**183**	*Antrodia camphorate* [298], *Gymnoascus reessii* [249], *Stereum hirsutum* [17]	**cytotoxic assay** (A549, HL-60, MCF-7, SMMC-7721, SW480, IC_50_ > 40 μM) [17], (KB, MCF-7, IC_50_ > 50 μM; NCI-H187, IC_50_ 22.6 μM; Vero, IC_50_ 43.8 μM) [249]
**184**	*Ganoderma resinaceum* [103]	**NO production inhibition assay** (56.37% inhibition at 50 μM) [103]
**185**	*Rhizoctonia solani* [300]	**antibacterial assay** (MIC 1 μg/mL against the Gram-positive bacteria *Bacillus subtilis*, *Staphylococcus aureus*, and MRSA; MIC 16 μg/mL against the yeast *Candida albicans*; MIC 64 μg/mL against the Gram-negative bacteria *Escherichia coli* and *Pseudomonas aeruginosa*) [300]
**186**	*Trichoderma asperellum* [301]	**antibacterial assay** (against *V. harveyi*, *V. splendidus*, and *P. citrea* with inhibitory zones of 10, 7.5, and 8.0 mm, respectively, at 50 μg/disk) [301], **antifungal assay** (MIC 12 μg/mL against *Glomerella cingulate*) [301]
**187**	*Antrodia camphorate* [310], *Arthrinium* sp. [314], *Aspergillus penicillioides* [205], *Colletotrichum* sp. [206], *Conocybe siliginea* [315], *Gymnascella dankaliensis* [303], *Neosartorya fennelliae, N. tsunodae* [316], *Pestalotiopsis* sp. [139], *Phomopsis* sp. [7], *Pleosporales* sp. [317], *Stereum hirsutum* [17], *Talaromyces purpurogenus* [318], *Talaromyces* sp. [255]	**cytotoxic assay** (P388, ED_50_ 2.2 μg/mL) [303], (A549, IC_50_ 4.4 μM; HL-60, IC_50_ 2.3 μM; MCF-7, IC_50_ 2.7 μM; SMMC-7721, IC_50_ 3.3 μM; SW480, IC_50_ 3.5 μM) [17], (K562, IC_50_ > 20 μM; ST26, IC_50_ 6.7 μM) [310], (A549, IC_50_ 21.3 μM; HL-60, IC_50_ 7.9 μM; MCF-7, IC_50_ 23.8 μM; SMMC-7721, IC_50_ > 40 μM; SW480, IC_50_ 14.2 μM) [318], **iNOS inhibitory assay** (IC_50_ 6.58 μM) [7], **NO production inhibition assay** (IC_50_ 13.04 μM) [7]
**188**	*Antrodia camphorate* [310], *Calvatia nipponica* [126], *Gymnascella dankaliensis* [303], *Stereum hirsutum* [17]	**cytotoxic assay** (P388, ED_50_ 2.8 μg/mL) [303], (MCF-7, IC_50_ > 100 μM) [126], (A549, IC_50_ 16.6 μM; HL-60, IC_50_ 15.6 μM; MCF-7, IC_50_ 17.2 μM; SMMC-7721, IC_50_ 16.3 μM; SW480, IC_50_ 17.3 μM) [17], (K562, IC_50_ 23.1 μM; ST26, IC_50_ 8.4 μM) [310]
**189**	*Periconia* sp. [304]	**antibacterial assay** (MIC 4 μg/mL against *Staphylococcus aureus*, MIC 32 μg/mL against *Enterococcus faecalis*; MIC > 100 μg/mL against all four Gram-negative bacteria tested) [304], **NO production inhibition assay** (IC_50_ > 40 μM) [304]
**190**	*Phomopsis* sp. [7]	**iNOS inhibitory assay** (IC_50_ 1.49 μM) [7], **NO production inhibition assay** (IC_50_ 4.65 μM) [7]
**191**	*Daldinia childiae* [306]	**cytotoxic assay** (MCF-7, SMMC-7721, SW480, IC_50_ > 40 μM) [306], **NO production inhibition assay** (IC_50_ 21.2 μM) [306]
**192**	*Pleurotus eryngii* [6]	**NO production inhibition assay** (IC_50_ 10.3 μM) [6]
**193**	*Pleurotus eryngii* [201]	**NO production inhibition assay** (NO produced 57.8% at 30 μM) [201]
**194**	*Pleurotus eryngii* [248]	**NO production inhibition assay** (IC_50_ 13.0 μM) [248]
**195**	*Tricholoma matsutake* [319], *Pleurotus eryngii* [248]	**AChE inhibitory assay** (62.8% inhibition at 50 μg/mL) [319], **NO production inhibition assay** (IC_50_ 25 μM) [248]
**196**	*Pleurotus eryngii* [248]	**NO production inhibition assay** (IC_50_ 23.6 μM) [248]
**197**	*Aspergillus flocculosus* [277]	**cytotoxic assay** (A549, HepG2, THP-1, IC_50_ > 80 μM) [277], **IL-6 immune-suppressive activity assay** (IC_50_ 22 μM) [277]
**198**	*Stropharia rugosoannulata* [307]	
**199**	*Stropharia rugosoannulata* [307]	
**200**	*Stropharia rugosoannulata* [307]	
**201**	*Cortinarius glaucopus* [195], *Stropharia rugosoannulata* [307]	
**202**	*Pleurotus eryngii* [201]	**cytotoxic assay** (RAW 264.7, IC_50_ > 30 μM) [201]
**203**	*Pleurotus eryngii* [201]	**cytotoxic assay** (RAW 264.7, IC_50_ > 30 μM) [201]
**204**	*Penicillium purpurogenum* [254]	**cytotoxic assay** (A549, IC_50_ 5.57 μM; HepG2, IC_50_ 4.44 μM; MCF-7, IC_50_ 5.98 μM) [254], **IL-6 immune-suppressive activity assay** (96.7% inhibition at 5 μg/mL) [254], **NO production inhibition assay** (70.7% inhibition at 5 μg/mL) [254]

## 10. Degraded Sterols

The progressive degradation of ergostane-type steroids through 5,6- and 9,10-oxidative cleavages leads to the loss of ring A and the formation of highly degraded sterols (Figure 22). The most common and best studied among them is demethylincisterol A_3_ (**206**). It demonstrated a potent activity against many cancer lines (Table 8). Cytotoxicity-guided investigation of Chinese mangrove *Rhizophora mucronata* endophytic *Pestalotiopsis* sp. yielded **206** as the most active compound with IC_50_ values reaching nanomolar order [139].

Luo et al. examined a collection of secondary metabolites of endophytic fungi in search for inhibitors of SH2 containing protein tyrosine phosphatase-2 (SHP2) [320]. The latter is an oncogenic phosphatase participating in many signaling cascades and identified as a potential therapeutic target for cancer. It was found that demethylincisterol A_3_ (**206**) inhibited the protein tyrosine phosphatase activity of SHP2 with an IC_50_ of 6.75 μg/mL. In comparison, sodium orthovanadate used as a positive control showed an IC_50_ value of 114 μg/mL.

Demethylincisterol A_3_ (**206**) revealed significant antibacterial activities against a number of pathogenic bacteria with MICs values ranging from 3.13 to 12.5 μM (MICs of the positive control ciprofloxacin varied from 0.78 to 1.56 μM) [321].

*Agrocybe chaxingu* extract was shown to have a very strong osteoclast suppression effect, useful in the prevention and control of osteoporosis. In search of the active components of this mushroom, Kawagishi et al. isolated a number of degraded sterols **208**–**212**, collectively called as chaxines [322,323]. The initially assigned 2′*S*,5′*S* stereochemistry of the A ring of chaxine B (**209**) was erroneous and was subsequently revised to 2′*R*,5′*S* [324,325]. Chaxines A-C were evaluated in the osteoclast-forming assay and were shown to suppress the rate of osteoclast formation with no cytotoxicity [322,323].

Chaxine C (**211**) was also isolated from traditional Chinese medicinal mushroom *Cordyceps jiangxiensis* under the name jiangxienone and showed promising results in inhibiting cancer cells [326]. Its IC_50_ values against A549 and SGC-7901 cells were six-fold lower than that of cisplatin.

Albocisterols A-C (**219**–**221**) isolated from cultures of *Antrodiella albocinnamomea* were tested for inhibitory activities against protein tyrosine phosphatase [327]. A mixture of compounds **220** and **221** exhibited significant activity with IC_50_ value of 1.1 μg/mL (IC_50_ 1.2 μg/mL for ursolic acid used as a positive control). The corresponding C-27 alcohol, albocisterol A (**219**), was inactive at 50 μg/mL.

**Table 8 molecules-27-02103-t008:** Sources and biological activity of fungal degraded sterols.

Compound	Fungal Source [Ref.]	Assays (Activity) [Ref.]
**205**	*Fusarium solani* [328]	**cytotoxic assay** (A549, HL-60, MCF-7, SMMC-7721, SW480, IC_50_ > 40 μM) [328], **COX-2 inhibitory assay** (IC_50_ > 20 μM) [328]
**206**	*Agrocybe chaxingu* [322], *Amauroderma amoiensis* [82], *Aspergillus* sp. [321], *Colletotrichum* sp. [206], *Gymnascella dankaliensis* [329], *Omphalia lapidescens* [16], *Pestalotiopsis* sp. [139,320], *Pleosporales* sp. [317], *Termitomyces microcarpus* [132], *Tricholoma imbricatum* [245], *Xylaria allantoidea* [330]	**AChE inhibitory assay** (<10% inhibition at 50 μg/mL) [82], **antibacterial assay** (MIC 12.5 μM against *S. aureus*, 3.13 μM against *S. epidermidis*, 3.13 μM against *B. cereus*) [321], **cytotoxic assay** (A549, IC_50_ 11.14 nM; Hela, IC_50_ 0.17 nM; HepG2, IC_50_ 14.16 nM) [139],(A549, IC_50_ 27.2 μM; HL-60, IC_50_ 18.1 μM; K562, IC_50_ 13.6 μM; MCF-7, IC_50_ 10.9 μM; SMMC-7721, IC_50_ 21.7 μM; SW480, IC_50_ 19.2 μM) [245], (GES-1, IC_50_ 7.81 μM; HGC-27, IC_50_ 51.16 μM; MDA-MB-231, IC_50_ 16.48 μM) [16], (HeLa, IC_50_ 2.24 μg/mL; HCT-116, IC_50_ 2.51 μg/mL; HT-29, IC_50_ 3.50 μg/mL; MCF-7, IC_50_ 3.77 μg/mL; Vero, IC_50_ 3.65 μg/mL) [330], (P388, ED_50_ 1.0 μg/mL) [329], **osteoclast differentiation assay** (at 4.8 μM suppressed the rate of osteoclast formation to 55%) [322], **protein tyrosine phosphatase assay** (IC_50_ 6.75 μg/mL) [320]
**207**	*Amauroderma amoiensis* [82], *Armillariella tabescens* [170], *Aspergillus aculeatinus* [331], *Aspergillus* sp. [332], *Pyropolyporus fomentarius* [333], *Tricholoma imbricatum* [245]	**AChE inhibitory assay** (46.3% inhibition at 50 μg/mL) [82], **cytotoxic assay** (A549, IC_50_ 7.1 μM; HL-60, IC_50_ 22.1 μM; K562, IC_50_ 17.1 μM; MCF-7, IC_50_ 18.9 μM; SMMC-7721, IC_50_ 19.3 μM; SW480, IC_50_ 16.7 μM) [245], (A549, IC_50_ 18.2 μM; HL-60, IC_50_ 23.9 μM; K562, IC_50_ > 40 μM; MCF-7, IC_50_ 16.9 μM; SMMC-7721, IC_50_ 27.3 μM; SW480, IC_50_ >40 μM) [333], **NO production inhibition assay** (IC_50_ 36.48 μM) [170]
**208**	*Agrocybe chaxingu* [322]	**osteoclast differentiation assay** (at 4.8 μM suppressed the rate of osteoclast formation to 6.7%) [322]
**209**	*Agrocybe chaxingu* [323]	**osteoclast differentiation assay** (at 3.1 μg/mL suppressed the rate of osteoclast formation to 66%) [323]
**210**	*Agrocybe chaxingu* [323]	
**211**	*Agrocybe chaxingu* [323], *Cordyceps jiangxiensis* [326], *Tricholoma imbricatum* [245], *Xylaria allantoidea* [330]	**cytotoxic assay** (A549, IC_50_ 7.9 μM; MCF-7, IC_50_ 10.2 μM) [245], (HeLa, IC_50_ 50.17 μg/mL; Vero, IC_50_ 76.57 μg/mL) [330], (A549, IC_50_ 2.93 μM; SGC-7901, IC_50_ 1.38 μM) [326], **osteoclast differentiation assay** (at 3.1 μg/mL suppressed the rate of osteoclast formation to 0%) [323]
**212**	*Agrocybe chaxingu* [323]	
**213**	*Hericium alpestre* [334]	**cytotoxic assay** (A549, IC_50_ 71.1 μM; HeLa, IC_50_ 69.6 μM; HT-29, IC_50_ 54.8 μM) [334]
**214**	*Antrodia camphorate* [335]	**cytotoxic assay** (A-2058, IC_50_ 31.1 μM; B16F10, IC_50_ 26.69 μM; Huh-7, IC_50_ 43.03 μM; MCF-7, IC_50_ 77.59 μM) [335]
**215**	*Ganoderma capense* [8]	**cytotoxic assay** (BGC823, Daoy, HCT116, HepG2, NCI-H1650, IC_50_ > 50 μM) [8]
**216**	*Ganoderma sinense* [220]	**cytotoxic assay** (SW1990, Vero, IC_50_ > 100 μM) [220]
**217**	*Daedaleopsis tricolor* [336]	**cytotoxic assay** (A-549, HL-60, MCF-7, SMMC-7721, SW480, IC_50_ > 40 μM) [336]
**218**	*Lenzites betulinus* [337]	**PTP1B inhibitory activity assay** (IC_50_ 21.5 μg/mL) [337]
**219**	*Antrodiella albocinnamomea* [327]	**PTP1B inhibitory activity assay** (no activity against DPP-IV and PTP1B at 50 μg/mL) [327]
**220**	*Antrodiella albocinnamomea* [327]	**PTP1B inhibitory activity assay** (IC_50_ 1.1 μg/mL in a mixture with **10**–**46**) [327]
**221**	*Antrodiella albocinnamomea* [327]	**PTP1B inhibitory activity assay** (IC_50_ 1.1 μg/mL in a mixture with **10**–**45**) [327]
**222**	*Phomopsis tersa* [338]	**cytotoxic assay** (A549, HepG2, MCF-7, SF-268, IC_50_ > 100 μM) [338]
**223**	*Tricholoma matsutake* [319]	**AChE inhibitory assay** (40.3% inhibition at 50 μg/mL) [319]

## 11. Conclusions

Fungi have been a traditional object of human practical interest throughout history. At first this was due to the nutritional value of mushrooms. Currently, fungi are attracting special attention as a source of a large number of biologically active compounds belonging to different classes: polyketides, terpenoids, peptides, alkaloids, etc., [339]. A wide variety of fungi secondary metabolites, their low content in natural material and the complexity of structural identification have led to the rapid development of research in this area only in the last two–three decades through the use of highly efficient methods of instrumental analysis and separation of complex natural compositions. A special place among fungi constituents is occupied by the metabolic products of ergosterol, the most important fungal sterol. Many of them are discussed in this review and some appear promising as leads for new medicines. At the same time, it is obvious that the described results not only characterize the achieved high level of research in this area, but also indicate directions for further scientific search, which is necessary for a better understanding of the content of the fungal metabolome and will allow revealing more fully the possibilities of practical use of its components in human healthcare.

## Figures and Tables

**Figure 1 molecules-27-02103-f001:**
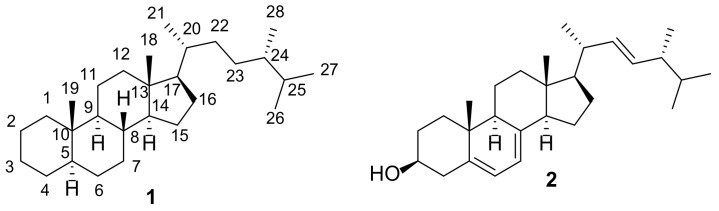
5α-Ergostane skeleton **1** and structure of ergosterol (**2**).

**Figure 2 molecules-27-02103-f002:**
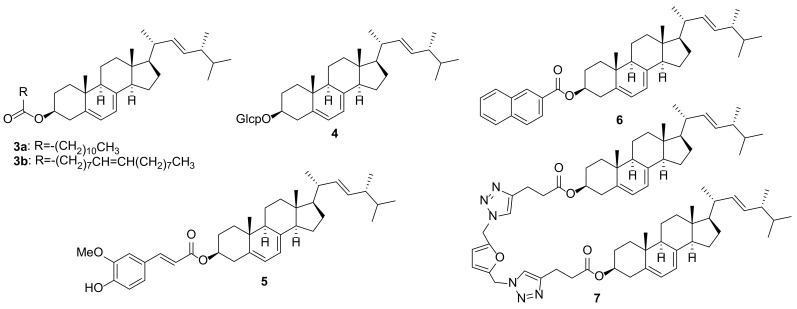
Structures of ergosterol *O*-derivatives.

**Figure 3 molecules-27-02103-f003:**
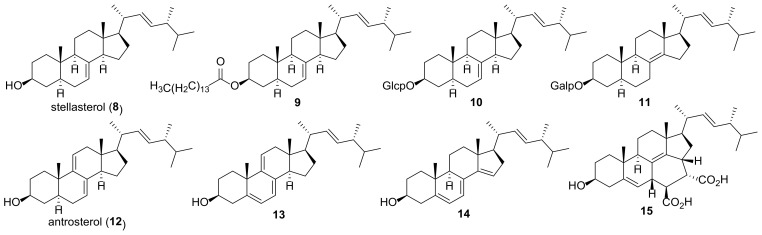
Structures of some fungal sterols and their derivatives.

**Figure 4 molecules-27-02103-f004:**
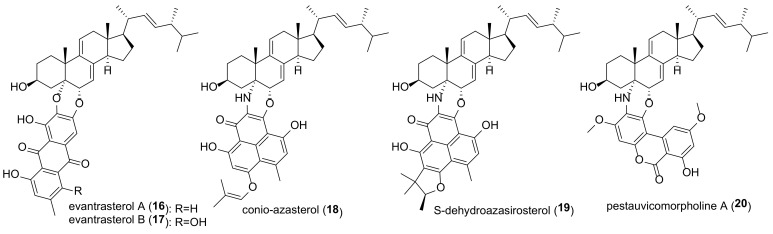
Structures of natural hybrids of 9-dehydroergosterol with polyketides.

**Figure 5 molecules-27-02103-f005:**
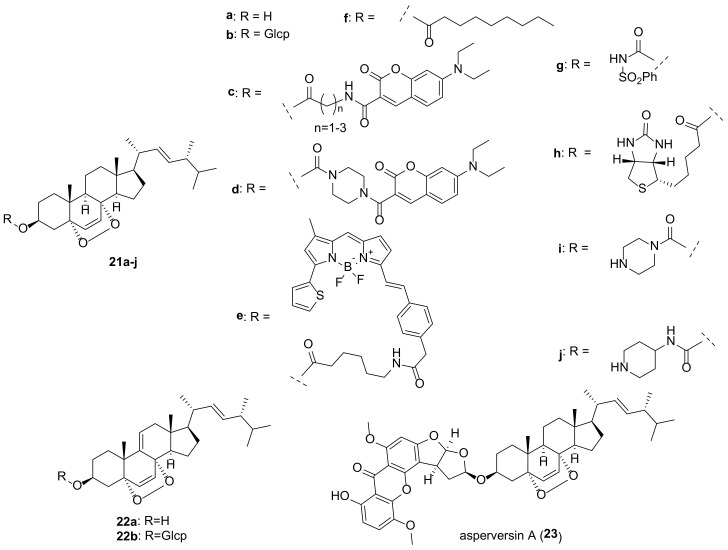
Structures of fungal 5α,8α-endoperoxides and their *O*-derivatives.

**Figure 6 molecules-27-02103-f006:**
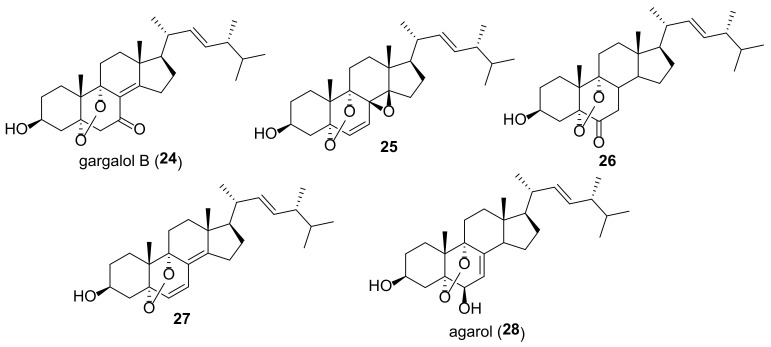
Structures of fungal 5α,9α-endoperoxides.

**Figure 7 molecules-27-02103-f007:**
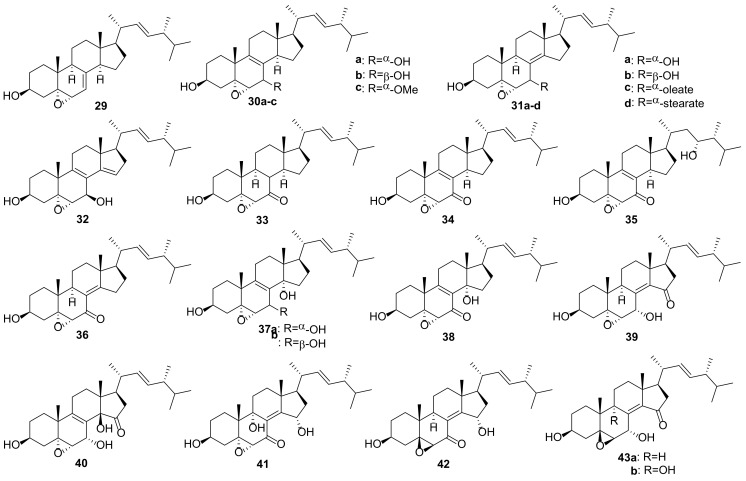
Structures of fungal 5,6-epoxides.

**Figure 8 molecules-27-02103-f008:**
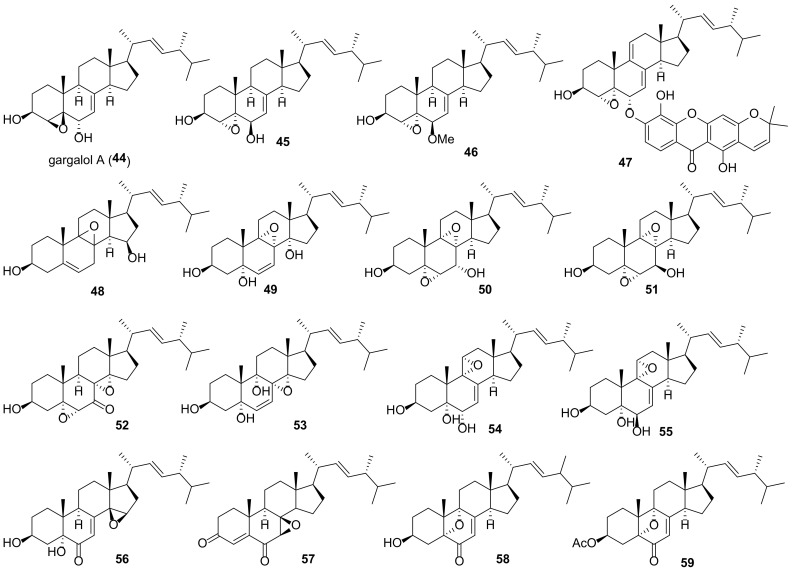
Structures of other fungal epoxides.

**Figure 9 molecules-27-02103-f009:**
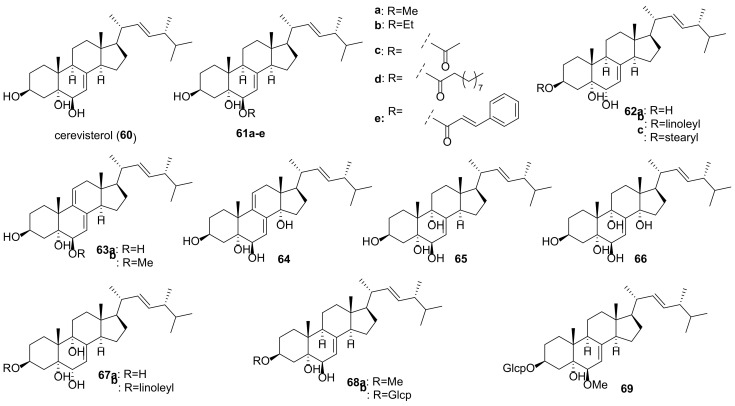
Structures of fungal steroids with a 5α,6-diol fragment and their *O*-derivatives.

**Figure 10 molecules-27-02103-f010:**
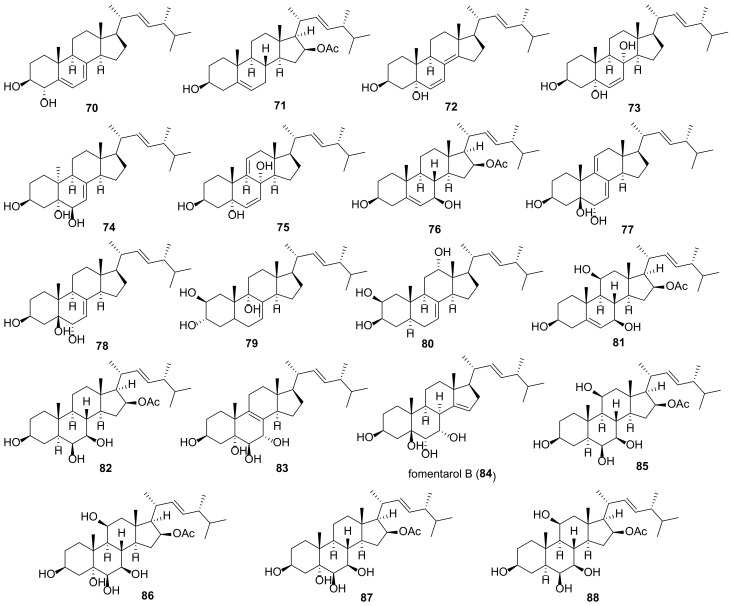
Structures of other fungal polyols.

**Figure 11 molecules-27-02103-f011:**
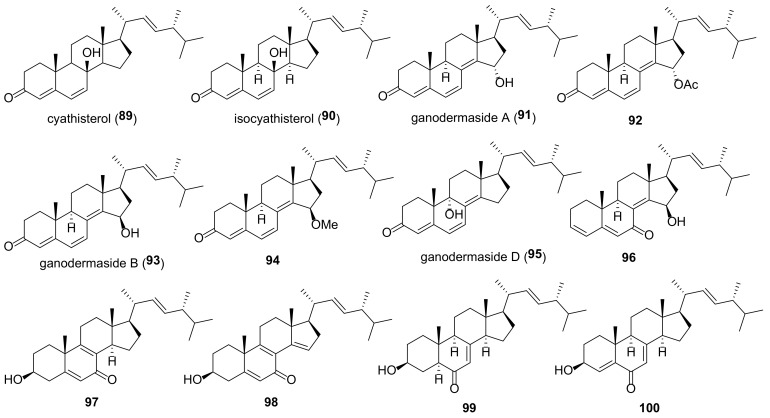
Structures of fungal hydroxyketones with two functional groups.

**Figure 12 molecules-27-02103-f012:**
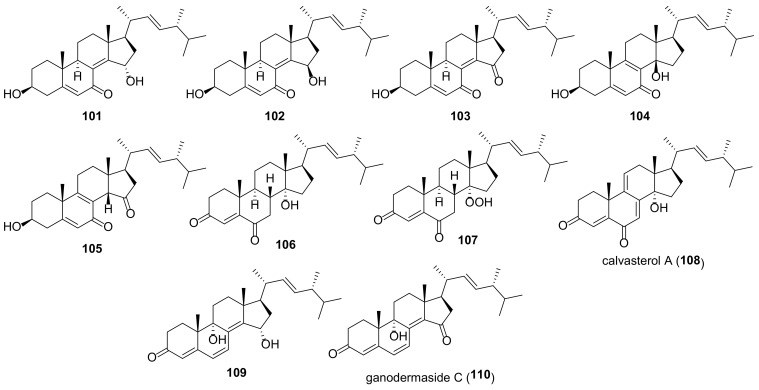
Structures of fungal hydroxyketones with three functional groups.

**Figure 13 molecules-27-02103-f013:**
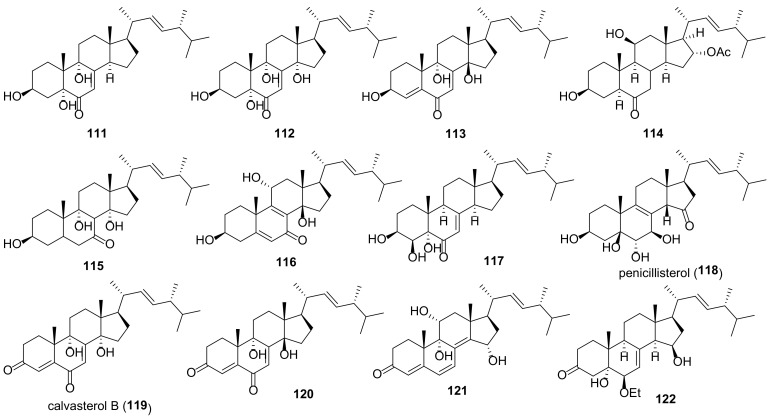
Structures of fungal hydroxyketones with four or more functional groups.

**Figure 14 molecules-27-02103-f014:**
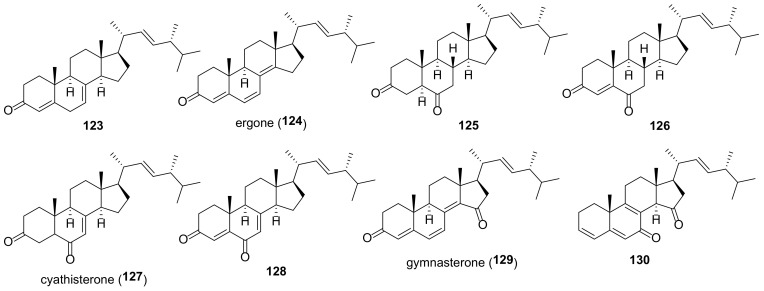
Structures of fungal ketones.

**Figure 15 molecules-27-02103-f015:**
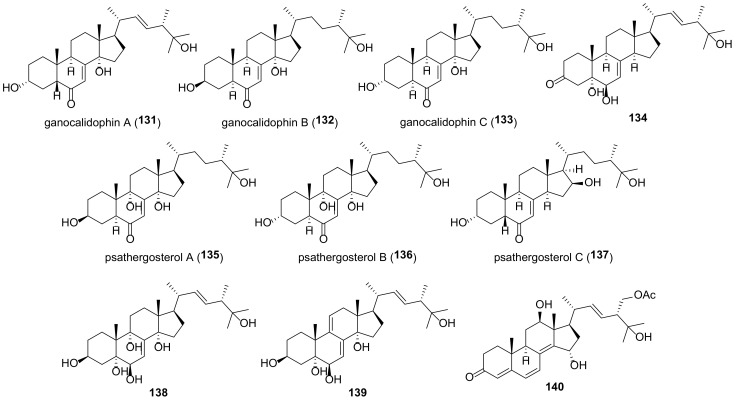
Structures of fungal 25-hydroxy steroids.

**Figure 16 molecules-27-02103-f016:**
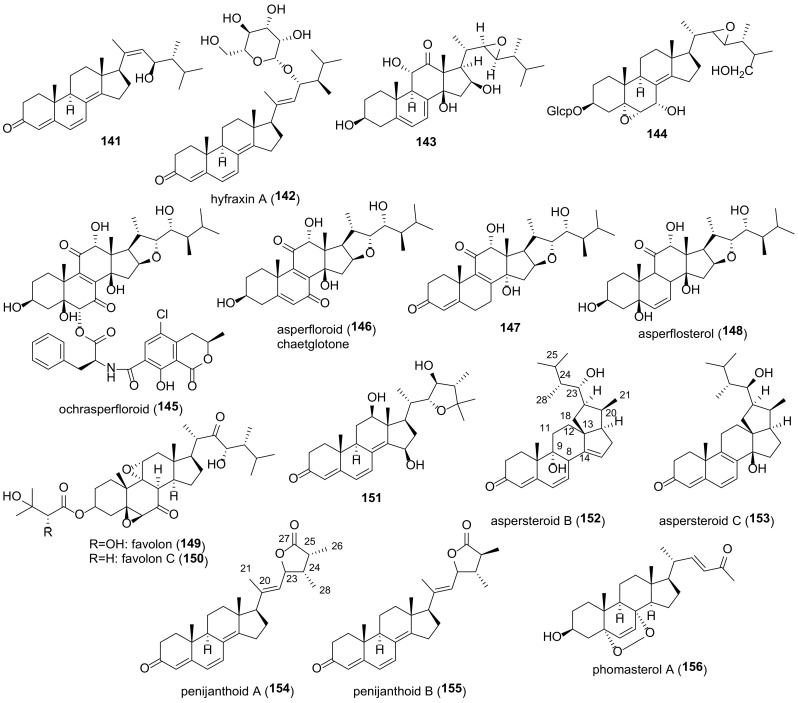
Structures of steroids with a transformed side chain.

**Figure 17 molecules-27-02103-f017:**
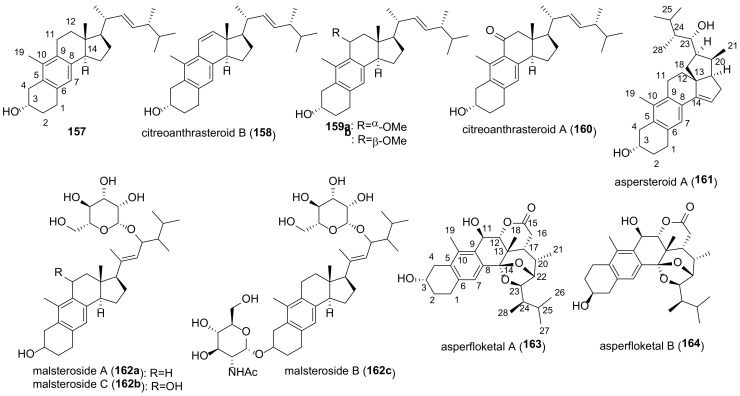
Structures of 1(10→6)abeo-ergostane-type steroids.

**Figure 18 molecules-27-02103-f018:**
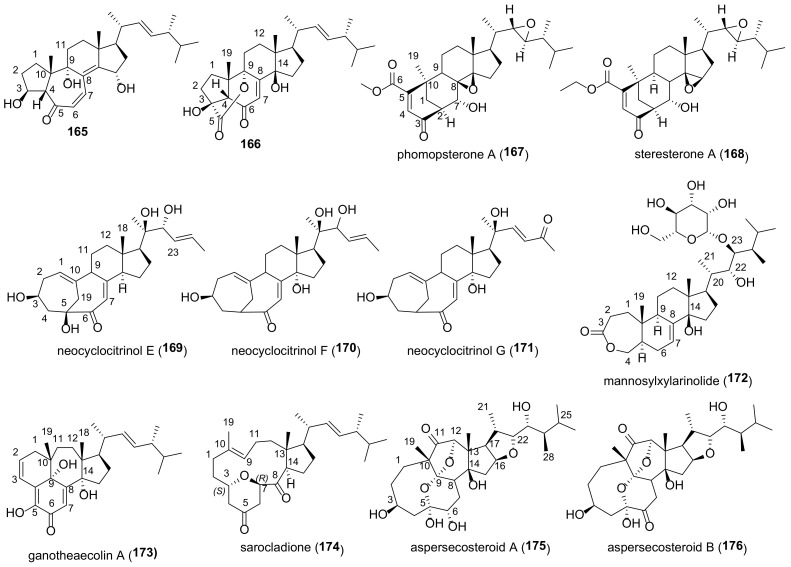
Structures of ergostanes with a rearranged A-ring.

**Figure 19 molecules-27-02103-f019:**
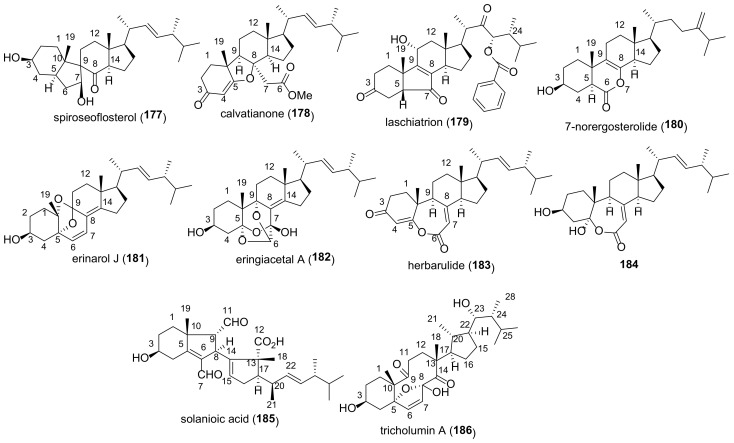
Structures of ergostanes with a rearranged B-ring.

**Figure 20 molecules-27-02103-f020:**
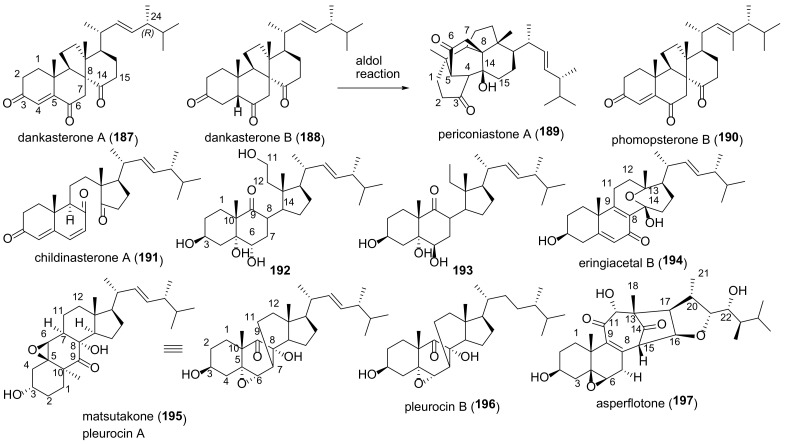
Structures of ergostanes with a rearranged C-ring.

**Figure 21 molecules-27-02103-f021:**
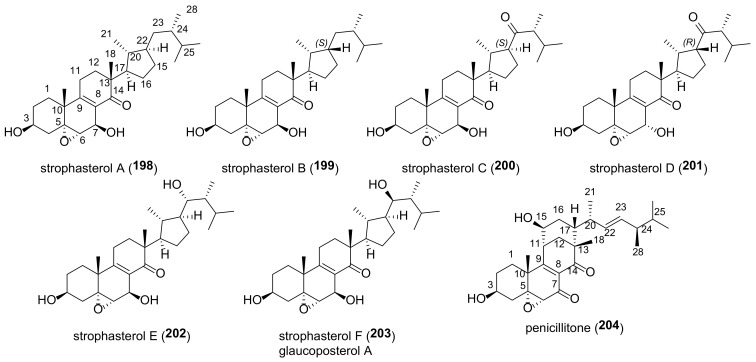
Structures of ergostanes with a rearranged D-ring.

**Figure 22 molecules-27-02103-f022:**
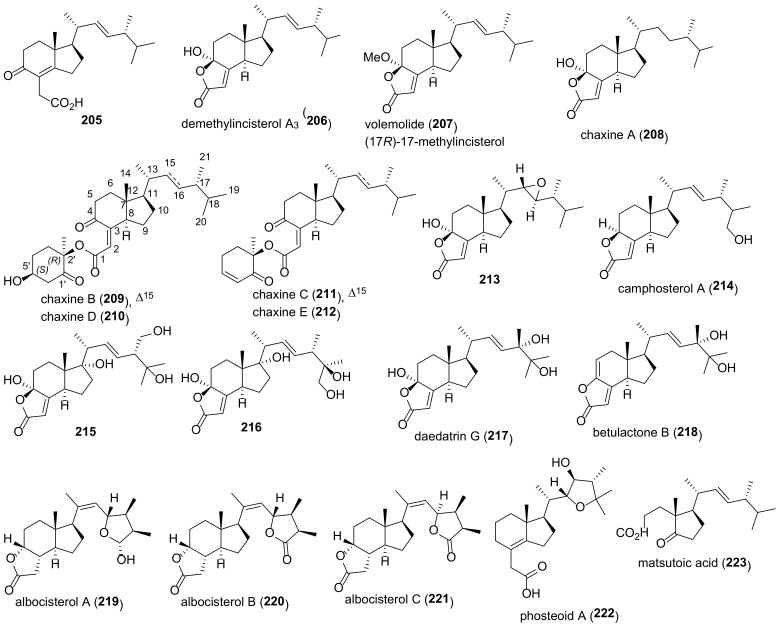
Structures of degraded sterols.

**Table 5 molecules-27-02103-t005:** Sources and biological activity of fungal ketones.

Compound	Fungal Source [Ref.]	Assays (Activity) [Ref.]
**123**	*Gymnoascus reessii* [249]	**antimalarial assay** (IC_50_ 3.3 μg/mL against *Plasmodium falciparum*) [249], **cytotoxic assay** (KB, IC_50_ 32.5 μM; MCF-7, IC_50_ > 50 μM; NCI-H187, IC_50_ 16.3 μM; Vero, IC_50_ 17.0 μM) [249]
**124**	*Antrodia cinnamomea* [263], *Aspergillus penicillioides* [205], *A. ustus* [231], *Colletotrichum* sp. [190], *Cortinarius xiphidipus* [85], *Fulviformes fastuosus* [264], *Ganoderma sinense* [220,233], *Gymnoascus reessii* [249], *Hygrophorus russula* [183], *Lentinus polychrous* [225], *Leucocalocybe mongolica* [210], *Mahonia fortune* [265], *Nigrospora sphaerica* [104], *Phellinus pini* [90], *Pleurotus tuber-regium* [228], *Polyporus umbellatus* [266,267], *Talaromyces* sp. [268], *Xylaria* sp. [259]	**antibacterial assay** (MIC 16 μg/mL against *Edwardsiella tarda* and *Micrococcus luteus*) [205], **antimalarial assay** (IC_50_ 4.5 μg/mL against *Plasmodium falciparum*) [249], **cytotoxic assay** (A549, IC_50_ 98.56 μM; HeLa, IC_50_ 53.19 μM; HepG2, IC_50_ 34.02 μM; MCF-7, IC_50_ 45.92 μM) [210], (HepG2, IC_50_ 68.32 μM; RD, IC_50_ 1.49 μM) [264], (LNCap, IC_50_ 34.7 μM; MCF-7, IC_50_ 57.5 μM; N2A, IC_50_ 20.8 μM; Saos-2, IC_50_ 27.8 μM) [85], (KB, IC_50_ 48.1 μM; NCI-H187, IC_50_ 58.8 μM) [269], (HL60, IC_50_ 30 μM; K562, IC_50_ 350 μM) [104], (KB, IC_50_ 40.9 μM; MCF-7, IC_50_ > 50 μM; NCI-H187, IC_50_ 47.9 μM; Vero, IC_50_ 49.2 μM) [249], (MDA-MB-231, IC_50_ 33 μM) [268], (A549, IC_50_ 18.8 μg/mL; XF498, IC_50_ 24.6 μg/mL) [183], (AGS, IC_50_ 56.1 μM; Hela229, IC_50_ 67 μM; Hep3B, IC_50_ 12.7 μM; HT-29, IC_50_ 18.4 μM;) [267], (HepG2, IC_50_ 10 μM) [270], (LU-1, IC_50_ 10.21 μg/mL) [271], **NO production inhibition assay** (IC_50_ 28.96 μM) [259], (IC_50_ 29.7 μM) [90]
**125**	*Stereum hirsutum* [17]	**cytotoxic assay** (A549, MCF-7, SMMC-7721, SW480, IC_50_ > 40 μM; HL-60, IC_50_ 34.3 μM) [17]
**126**	*Stereum hirsutum* [17], *Xerula furfuracea* [10]	**cytotoxic assay** (A549, HL-60, MCF-7, SMMC-7721, SW480, IC_50_ > 40 μM) [17]
**127**	*Apiospora montagnei* [269], *Gymnoascus reessii* [249]	**cytotoxic assay** (NCI-H187, IC_50_ 14.8 μM) [269], (KB, MCF-7, NCI-H187, Vero, IC_50_ > 50 μM) [249]
**128**	*Polyporus ellisii* [198]	**HNE inhibitory assay** (IC_50_ 55.2 μM) [198]
**129**	*Phomopsis* sp. [202], *Polyporus ellisii* [184], *Talaromyces stipitatus* [204]	**α-glucosidase inhibition assay** (IC_50_ > 100 μM) [202], **cytotoxic assay** (Hep3B, IC_50_ 36.27 μM; HepG2, IC_50_ 36.51 μM) [204]
**130**	*Tricholoma imbricatum* [245]	**cytotoxic assay** (A549, IC_50_ 22.8 μM; SMMC-7721, IC_50_ 19.5 μM) [245]

## Data Availability

Not applicable.

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
