# Peer review of "Structure and Biological Activity of Ergostane-Type Steroids from Fungi"

_molecules, 2022, doi:10.3390/molecules27072103_

Round 1

Reviewer 1 Report

This manuscript reviewed the bioactive secondary metabolites, ergosterol, from fungal sources within the past 10 years. This review will provide valuable information for other researchers. If possible, the author could include some information on the glycosides of ergosterol or add some discussion on the SARs. Overall, it is a well written review and is acceptable after some minor revisions.

Author Response

> If possible, the author could include some information on the glycosides of ergosterol...

The following paragraph has been added: "

The glucopyranosyl derivative 4 showed slightly higher activity in inhibiting LPS-induced NO production than ergosterol (1) (IC50 16.6 and 14.3 μM, respectively) [69]. On the other hand, COX-1 enzyme inhibitory activity of 4 was weaker compared with that of the aglycone 1 [70]."

Reviewer 2 Report

Vladimir N. Zhabinskii’s manuscript is a comprehensive review on the diverse structures and bioactivities of fungal ergosterols, which worth reading. However, this article still has some to improve.

  1. Mushrooms are a kind of fungi, thus there are some distinctions between mushrooms and fungi the fungi. It is better for authors to think over the topic of this manuscript, then check the title, abstract and introduction.
  2. Page 1 Line 15: ‘bioactivity’ → ‘bioactivities’
  3. There are some references’ numbers missing. For instance, [18] in Page 1, [55, 56] in Page 3, [85, 86] in Page 4. Please check all the references’ numbers throughout the whole manuscript carefully.
  4. Page 1 Line 37: ‘structure’ → ‘structure s’
  5. Please mark the configurations of C-8, 9, 14 and 17 in 5α-ergostane skeleton in Figure 1.
  6. Page 2 Line 68: Please check the unit in the sentence ‘…IC50 value of 40 μM/mL’.
  7. Page 3 Line 99: ‘[52, 53’ → ‘[52, 53]
  8. Please remove the compounds’ names in Figure 3 for consistency. And please check other figures carefully.
  9. Page 5 Line 198: ‘]]. It has been…’ → ‘It has been…’
  10. Page 5 Line 206: Please cite a reference after ‘…Antrodia camphorate’.
  11. Page 12 Lines 388 and 390: ‘M. tuberculosis’ → ‘M. tuberculosis
  12. Page 13 Line 440: ‘4 cancer lines’ → ‘four cancer lines’
  13. Page 24 Line 561: ‘63 compounds’ → ‘sixty-three compounds’
  14. Why the numbering for compound aspersteroid A is 161, but for the aforementioned malsterosides is 162?
  15. Page 34 Line 712: ‘4 cancer lines’ → ‘four cancer lines’
  16. Table 8: ‘PTP1Bc’ → ‘PTP1B
  17. Please add ‘MEG2, DPPIV and PTP1B’ in the list of Abbreviations.
  18. Please use the abbreviation for the journal name ‘Evidence-based 984 Complementary and Alternative Medicine’ in Ref. [2].
  19. Please revise the page number in Ref. [315]: ‘375–1’ → ‘375’

Author Response

>1. Mushrooms are a kind of fungi, thus there are some distinctions between mushrooms and fungi the fungi. It is better for authors to think over the topic of this manuscript, then check the title, abstract and introduction.

Abstract has been modified: "Mushrooms are known not only for their taste but also for beneficial effects on health attributed to plethora of constituents. All mushrooms belong to the kingdom of fungi, which also includes yeasts and molds. Each year, hundreds of new metabolites of the main fungal sterol, ergosterol, are isolated from fungal sources. "

>2. Page 1 Line 15: ‘bioactivity’ → ‘bioactivities’ -> Corrected.

>3. There are some references’ numbers missing. For instance, [18] in Page 1, [55, 56] in Page 3, [85, 86] in Page 4. Please check all the references’ numbers throughout the whole manuscript carefully.

This is a technical issue. The manuscript was uploaded as a .docx file containing EndNote fields (ensuring correct numbering of all references). The submission system generated own .pdf file based on the uploaded .docx file. Some fragments of the initial text were lost during this .docx->.pdf transformation. Evidently, the EndNote fields should have been removed before the submission (this has been done in the revision).

>4. Page 1 Line 37: ‘structure’ → ‘structure s’ ->Corrected.

>5. Please mark the configurations of C-8, 9, 14 and 17 in 5α-ergostane skeleton in Figure 1. -> Corrected.

>6. Page 2 Line 68: Please check the unit in the sentence ‘…IC50 value of 40 μM/mL’. -> The information corresponds to that published in the cited article.

>7. Page 3 Line 99: ‘[52, 53’ → ‘[52, 53]’ -> The same as in the Note 3 (technical issue).

>8. Please remove the compounds’ names in Figure 3 for consistency. And please check other figures carefully -> Many (but not all) compounds have trivial names which are well known to those working in this field. We would like to keep all trivial names, as it is a valuable additional information for specialists.

>9. Page 5 Line 198: ‘]]. It has been…’ → ‘It has been…’ -> The same as in the Note 3 (technical issue).

>10. Page 5 Line 206: Please cite a reference after ‘…Antrodia camphorate’. -> The same as in the Note 3 (technical issue).

>11. Page 12 Lines 388 and 390: ‘M. tuberculosis’ → ‘M. tuberculosis’ -> Corrected.

>12. Page 13 Line 440: ‘4 cancer lines’ → ‘four cancer lines’ -> Corrected.

>13. Page 24 Line 561: ‘63 compounds’ → ‘sixty-three compounds’ -> Corrected.

>14. Why the numbering for compound aspersteroid A is 161, but for the aforementioned malsterosides is 162? -> The relevant paragraphs are swapped .

>15. Page 34 Line 712: ‘4 cancer lines’ → ‘four cancer lines’ -> Corrected.

>16. Table 8: ‘PTP1Bc’ → ‘PTP1B’ -> Coprrected.

>17. Please add ‘MEG2, DPPIV and PTP1B’ in the list of Abbreviations. -> Corrected.

>18. Please use the abbreviation for the journal name ‘Evidence-based 984 Complementary and Alternative Medicine’ in Ref. [2] -> Corrected.

>19. Please revise the page number in Ref. [315]: ‘375–1’ → ‘375’ -> Corrected.

Reviewer 3 Report

This manuscript is a review article focusing on the structure and biological activity of fungal metabolites of ergosterol. The authors aim to review literature mostly from the last ten years. Of the 338 references quoted, there are 236 from 2015 to present and 70 in the last two years (2020-2022).  As a review, there is no experimental protocol per se to be considered nor the validity of data analysis. However, at 69 pages (of which 23 are references), it is very long for such a specific topic.  

It is generally well written, comprehensive, and well organized. There are few, if any, typos or grammar issues. The discussion is very detailed which results in the manuscript being overly lengthy and therefore difficult to read. It covers some 223 different chemical structures that are organized according to common structural elements. There are lengthy tables describing the cytotoxicity (Table 1) and fungal source and biological activity (Tables 2 to 8) for each structural group of steroids. The text to some extent duplicates what is presented in the tables, which provides an opportunity to condense the manuscript somewhat.

If there is a criticism, it is that the manuscript is too lengthy to be easily read and while it thoroughly covers what is present in recent literature, the manuscript could be improved by condensing the text and summarizing the trends in biological activity that are so specifically outlined in the tables.

Specific comments

Page 5, lines 206-207: There is a spacing issue. Are Antrodia camphorate and Coprinus setulosus two different organisms? Was there supposed to be a reference for Antrodia camphorate?

References, pages 48 to 68  : There are spacing issues for references 9, 18, 19, 34, 40, 67, 95, 102, 121, 122, 131, 147, 158, 159, 170, 185, 188, 212, 217, 239, 244, 285, 326 and maybe more. This may be a justification issue but all the other references are fine.

Author Response

>If there is a criticism, it is that the manuscript is too lengthy to be easily read and while it thoroughly covers what is present in recent literature, the manuscript could be improved by condensing the text ...

This is a kind of art to find the right balance between brevity and completeness. Perhaps the present manuscript is somewhat long, but between brevity and completeness, we would like to choose the latter.

>Page 5, lines 206-207: There is a spacing issue. Are Antrodia camphorate and Coprinus setulosus two different organisms? Was there supposed to be a reference for Antrodia camphorate?

This was a technical mistake resulting from .docx->.pdf transformation of texts containing EndNote fields. In the revision this problem is solved.

>References, pages 48 to 68  : There are spacing issues for references 9, 18, 19, 34, 40, 67, 95, 102, 121, 122, 131, 147, 158, 159, 170, 185, 188, 212, 217, 239, 244, 285, 326 and maybe more. This may be a justification issue but all the other references are fine.

This is indeed a justification issue.

Round 2

Reviewer 2 Report

The comments has been addressed and the paper is revised accordingly. No further suggestion from me.